# A scope of prebiotic neat reaction conditions and the mechanism of urea-assisted phosphorylations of alcohols

Anastasiia Shvetsova[1,2], Lynda Merzoud[3], Augustin Lopez[1], Elodie Fromentin[1], Anne Baudouin[1], Henry Chermette [3], Isabelle Daniel [2], Michele Fiore [1] & Peter Strazewski [1] ✉

Two theories of the origin of life on Earth, one located in the Hadean seafloor, the other on the surface of subaerial landmasses and basins, need reconcilement. Natural energy flows moulded seafloors to locally sustain chemical reaction networks reminiscent of metabolism. Subaerial hot milieus gathered organic phosphates to provide robust compartments from phospholipids and heredity from nucleic acid replication and translation. Here, we report on the efficiency and product distribution of the phosphorylation of twelve model alcohols reacting separately, and in selected combinations, all under the same chemically activated conditions, mostly as neat hot solid-liquid mixtures. We afford proof for the dominant reaction mechanism and indicate how prebiotic seafloor and subaerial systems could profit from one another through atmospheric and tidal exchange of organic material.

Phosphate takes an indispensable role in all three kingdoms of life, Bacteria, Archaea and Eukaryota, and currently represents as much as five gigatons of phosphorous in the biosphere[1,2]. Before the advent of enzymatic catalysis and the emergence of primary metabolic reaction networks[3], and also before organic phosphate esters could accumulate on the early Earth and serve as available phosphate sources themselves[4], they must have been first formed under abiotic conditions from organic precursor molecules and dissolved phosphate minerals[5–10]. The thermodynamics in water are challenging, since the endergonic nature of the phosphorylation of typical organic precursors with soluble inorganic orthophosphate ($P_i$) requires chemical energy[11,12].

Out of many proposed suitable prebiotic activation conditions, relatively long-lasting non-volatile reservoirs of cyanate ($NCO^-$) and inorganic carbamoyl phosphate ($CP_i$), from $NCO^- + O_2P(OH)_2^- \rightleftarrows H_2NCOOPO_3^{2-}$ being present in inorganic phosphate-urea blends, seem most attractive for their reactivity with amines as N-carbamoylation agents[13,14] and as O-phosphorylation agents for hydroxyl-bearing organic precursors[15]. However, the lately growing

number of different prebiotic dehydrating conditions[16–19] proposed to have furnished crucial organic phosphate esters shortly before the first living cells have emerged in the late Hadean and early Archean eras[20,21], leave at present the scientific community with a very large range of possibilities that lack thus far appropriate experimental comparison.

The Earth's first solid subaerial locations amenable to accumulate on their surface significant amounts of organic material were likely recurrently wetted with rivulets from hot springs[22] or ocean seawater spawning higher waves[20] and tides more violently rising and retrieving with the rhythm of the moon that orbited closer to Earth and faster than today[23,24]. Rocky surfaces were thus provided with water containing mostly hydrogen- and oxygen-bearing organics. Such compounds could accumulate over longer periods of time, unless they were too volatile, together with a lower abundance of non-volatile N-heterocycles, amino acids and soluble inorganic phosphate sources[25–28]. Regarding reduced-nitrogen compounds, there is a faint possibility that solid ammonia-formaldehyde stockpiles might have been blended with inorganic phosphide, corroding to phosphites and phosphates at locations where large deposits of

[1]Institut de Chimie et Biochimie Moléculaires et Supramoléculaires, UMR5246, CNRS, Université Claude Bernard Lyon 1, Villeurbanne, France. [2]Laboratoire de Géologie de Lyon–Terre, Planètes et Environnement, UMR5276, CNRS, Université Claude Bernard Lyon 1, Villeurbanne, France. [3]Institut de Chimie Analytique, UMR5280, CNRS, Université Claude Bernard Lyon 1, Villeurbanne, France. ✉e-mail: strazewski@univ-lyon1.fr

hexamethylenetetramine (sublimation at 280 °C and 1 atm) would be wetted, that had been delivered from asteroid impact[29] together with schreibersite[30,31]. Usually however, at daytime, shallow basins were periodically drying out, some likely within a few hours, consequently, electron-rich N-volatiles such as ammonia, methyl amine or hydrolytically unstable N-methyl isocyanide could barely amass or be "fixed" in less volatile and highly energetic deposits, such as inorganic amidophosphite for instance, or mono- and diamidophosphates[13,32–34]. With such a very early, amine-poor, mostly hot and dry scenario in mind, being limited in badly soluble inorganic phosphate, we wondered if the phosphorylation of important prebiotic organic precursors could have happened in the absence of liquid water, in neat reaction conditions.

Here, we show how the phosphorylation of the abiotic or biotic precursors **5–16** (Fig. 1) of membrane components, nucleotides and phosphoenol pyruvate occurs under mostly neat and hot conditions (minimum 60, mostly 115, maximum 130 °C) in the solid or mixed solid-liquid phase, without episodic or periodic wet-dry cycling[17,35], in the presence of an equimolar quantity of a pre-mixed (as if evaporated from aqueous) inorganic phosphate source $P_i$[28], $SP_i$[19,36], $PP_i$ or cTMP[37,38] (Fig. 1) and one of the most simple, moderately volatile compounds **1–4** bearing reduced-nitrogen that could have been very early, abiotically "fixed" from atmospheric $N_2$[21,39–44]. Our results demonstrate that the quantified efficiency of alcohol phosphorylation in neat conditions is particularly high for triols and diols (**5, 6** and **11–15**); it works best when assisted by urea (**2a**) or its hydrolytic precursor cyanamide (**1**), whereas mono-functional alcohols (**7–10, 16a**) are hardly phosphorylated in the same neat conditions, or not at all. We show how the different phosphate sources compare with one another, and confirm

that neat cTMP efficiently phosphorylates only **5** and **6** but not the nucleosides (**11–15**). We provide proof of the reaction mechanism of urea-assisted phosphorylations by quantum theoretical and experimental means; the latter, in a time-dependent fashion and with the help of reagents enriched with the stable isotopes $^{15}N$ (**2b, 3b, 4b**), $^{13}C$ (**2c**) and $^{18}O$ in inorganic orthophosphate [$^{18}O_4$]$P_i$. We present the molecular structure of a wide range of different phosphorylated organic products identified in part by additional blending the neat mixtures with glyceryl phosphate (**5a**), an equimolar mixture of decanoic, undecanoic, dodecanoic and tridecanoic acids (**7abcd**), one of the nucleoside-5′-monophosphates (NMP **11a–14a**), phosphoenol pyruvate (**16b**), or one of the amino acids L-alanine, L-valine or D-valine (**22abc**); see Supplementary Table 1 for a complete list. We conclude on how the different starting alcohols and other compounds, that could have been present in dried-out complex prebiotic mixtures, must have competed for inorganic phosphate sources available on the subaerial landmasses of the late Hadean era. Finally, we indicate how important and useful such arid environments could have been for the emergence of the first living cells forming under water at the ocean seafloor.

## Results

### Scope of neat phosphorylation conditions

Briefly, in a nut shell: out of all tested alcohols **5–16a** we have found that their blending with $P_i$ and cyanamide (**1**) or urea (**2**) in "dry" mixtures, the latter containing initially about 4 to 15 weight% residual water, were generally the most resourceful and robust neat phosphorylation conditions on the triols **5, 11–14** and diols **6** and **15** in providing some endurance by way of forming organic carbamates and

**Fig. 1 | Dehydrating agents (1, 2), liquid additives (3, 4), inorganic phosphate sources, and alcohols (5–16a) used in this work. 2a, 3a, 4a**: natural isotope-abundance; **2b, 3b, 4b**: enriched in $^{15}N$; **2c**: enriched in $^{13}C$. All alcohols and other compounds, except urea (**2abc**), the nucleosides (**11–15**) and the phosphate sources, are liquid at 80 to 130 °C. Known standard free enthalpies of hydrolysis (at 25 °C in water) are marked next to arrows (left) or added in brackets (middle), to be compared with $\triangle G^{\ominus} = -2.2$ kcal/mol for the hydrolysis of glyceryl phosphate, −3 to −5 kcal/mol for that of various sugar and nucleoside monophosphates, −7 to −12 kcal/mol for phosphoric anhydrides (di- and triphosphates), up to −14.8 kcal/mol for the hydrolysis of phosphoenolpyruvate[11]. * Urea[45,46,48] and formamide[93] (and, by analogy, other carboxamides) have been claimed in the literature to act as possible organo-catalysts in the phosphorylation of alcohols, cf. Fig. 4A and Supplementary Information (section 3.1).

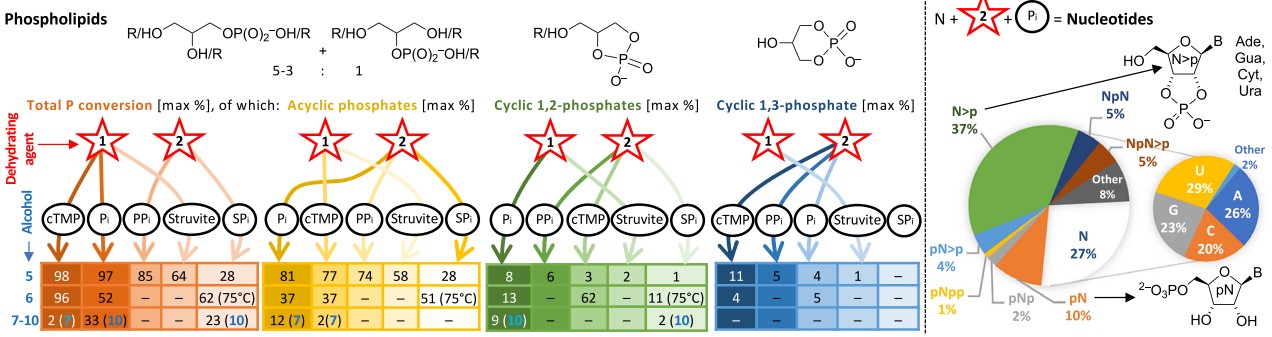

**Fig. 2 | Maximum percent conversion of $P_i$, cTMP, $PP_i$ and $SP_i$ after 5 days at 115 °C in neat conditions.** Coloured boxes: Highest % $P_i$ conversion (from $^{31}$P NMR peak integration) to phospholipid precursors from alcohols **5, 6** (R = palmitoyl) and **7–10** blended with the phosphate source (black circles or ovals) and the dehydrating agent **1** or **2** (red stars), 1 mol equivalent each, all measured reaction scales included, maximum values for each class of compounds (acyclic, cyclic 1,2 and cyclic 1,3). Pie charts (right): Summary of nucleotide distribution (larger) and nucleobase identity (smaller) after heating the neat ribonucleosides (N) at 115 °C, mol ratio **11:12:13:14:2**:$P_i$ 0.25:0.25:0.25:0.25:4:1 and 1 eq **3a** as liquidiser (from signal integration of $UV_{260 nm}$ using HPLC-HRMS). Other = non-canonical nucleobases (B) and nucleos(t)ides, see Supplementary Figs. 156 and 157.

cyclic carbonates in addition to phosphate esters; mostly acyclic primary phosphates for **5, 6** and **15**, predominantly 2′,3′-cyclic phosphates for **11–14**. Slightly acidic neat **2** + $P_i$ mixtures (better than **1** + $P_i$) with excess **2** were much more beneficial for **11–15** than alkaline neat mixtures, where $SP_i$, $PP_i$ and cTMP proved superior for **5** and **6** but did not phosphorylate **11–15** at all. The presence of L-valine suppressed somewhat the phosphorylation of **11–14** by **2** + $P_i$, and more so than D-valine, suggesting a tighter complexation between D-ribonucleosides and L-valine than D-valine. Heated neat blends of **2** + $P_i$ with **5, 7abcd** and **11–14** contained up to diglyceryl, dinucleosidyl and glyceryl-nucleosidyl phosphodiester derivatives (Supplementary Fig. 171). We were sometimes successful in observing vesicles upon rehydrating crude phosphorylation extracts containing MPG phosphates from heated neat **6** + **2** + $P_i$ (Supplementary Fig. 223). By contrast, all precursor molecules furnished with only one hydroxyl group (the monools **7, 8, 9, 10** and enol **16a**) were by far less efficiently phosphorylated in all tested conditions, or not at all (Fig. 2).

## Kinetics and mechanism of neat urea-assisted phosphorylation conditions

Given the importance of neat hot urea-phosphate blends, we investigated the kinetics of exemplary phosphorylation reactions taking place at 115 °C in the solid or mixed solid-liquid state. The transient presence of NCO⁻ in neat urea + $P_i$ was proven by the observation of [$^{14}$N,$^{15}$N]urea from the reaction time-dependent exchange between natural urea (**2a**) and [$^{15}$N$_2$]urea (**2b**) in hot neat mixtures of **5** + **2a** + **2b** + $P_i$ and **6** + **2a** + **2b** + $P_i$ (red filled circles in Fig. 3, see also Supplementary Figs. 47–53). We determined the apparent kinetic rate constants $k$ (Fig. 3) of the consumption of **2ab**, $P_i$ and the formation of well-quantified products in the first 4–8 h (major carbamates and cyclic carbonates).

We have also elucidated the chemical mechanism of this reaction by catching all gasses in specific traps—in the form of benzamide and benzoic acid (white zones in Fig. 4)—that escaped from neat mixtures containing inorganic phosphate that was highly enriched in the $^{18}$O isotope. We could thus dismiss the urea-catalysed phosphate dehydration pathway[45] and confirmed this by density functional theory calculations, where a dihydrogen phosphate anion was placed between three molecules of urea and, in a second calculation, dihydrogen phosphate was surrounded by three urea and two assisting water molecules, and then a methanol molecule was added (Fig. 4A, B and Supplementary Figs. 13–15). The calculated rate-limiting step on the urea-catalysed phosphate dehydration pathway (TS-1, elementary

step sequence E-A$_N$-E-A$_N$, cf. Fig. 4A) proved too high, both, in neat urea (denoted U3W0) and in urea containing water molecules in a ratio 3:2 per dihydrogenphosphate (U3W2). The dissociative phosphate activation pathway was calculated to be faster, where dihydrogen phosphate allows the first tautomerisation of one urea molecule to its zitterionic form (Int-1, elementary step sequence taut-E-taut-A$_N$-A$_N$-E-E, cf. Fig. 4B) either in the absence of water (for U3W0: rate-limiting direct tautomerisation shown in Fig. 4B) or in its presence through a proton-relay by an assisting water molecule (for U3W2: blue TS-1 in Supplementary Fig. 15). The water-relayed tautomerisation step gives way to the trajectory-determining transition state that follows it (value highlighted in bright yellow in Fig. 4B). This zitterionic urea tautomer is predisposed to eliminate ammonia giving isocyanic acid, thereafter, generating with $P_i$ inorganic carbamoyl phosphate (CP$_i$) on a relatively smooth U3W2 potential energy surface (Int-3 → Int-5). This hints at the important accelerating role of residual water molecules initially present in the neat urea + $P_i$ mixture, presumed to be crystal water. Both dissociative pathways U3W0 and U3W2 provide a good rationale for the time-dependent formation of carbamoylated and carbonylated organic products found experimentally, including those long-lasting carbonylated diglyceryl phosphodiester derivatives that began to appear after about 8 h at 115 °C (Fig. 4C).

## Discussion

Heating time-dependent selectivity of the urea-assisted phosphorylation of glycerol (**5**) and MPG (**6**): Unlike glycerol phosphorylations carried out at 120 °C in dimethyl formamide solution with a tenfold excess urea and 20 mol% ammonium dihydrogenphosphate[16], the organophosphates obtained here were predominantly acyclic phosphoric mono- and -diesters, where the primary alcohols were always more abundantly phosphorylated than the secondary (usually ~ 5:1 RCH$_2$OPO$_3$$^{2-}$/R$_2$CHOPO$_3$$^{2-}$, cf. Supplementary Fig. 19 and 21C), and cyclic phosphates eventually appeared too, both, 5-membered rings (after 30 min) and 6-membered rings (after about 8 h) accompanied by the formation of primary carbamates and cyclic carbonates at the remaining alcohol groups (Fig. 3). We have recurrently observed and characterised in the DMSO extracts the formation of diglyceryl phosphodiester compounds starting to appear after 8 h at 115 °C (Fig. 4C and Supplementary Figs. 45 and 46) reaching peak abundance after only 96 h (Supplementary Table 12, columns 3 and 4). This has an important bearing on the prebiotic appearance of precursors of phospholipids having phosphoglyceryl (PG) and phosphoglyceryl-(cyclic-)phosphate (PGP, PGcP) polar headgroups widely used in

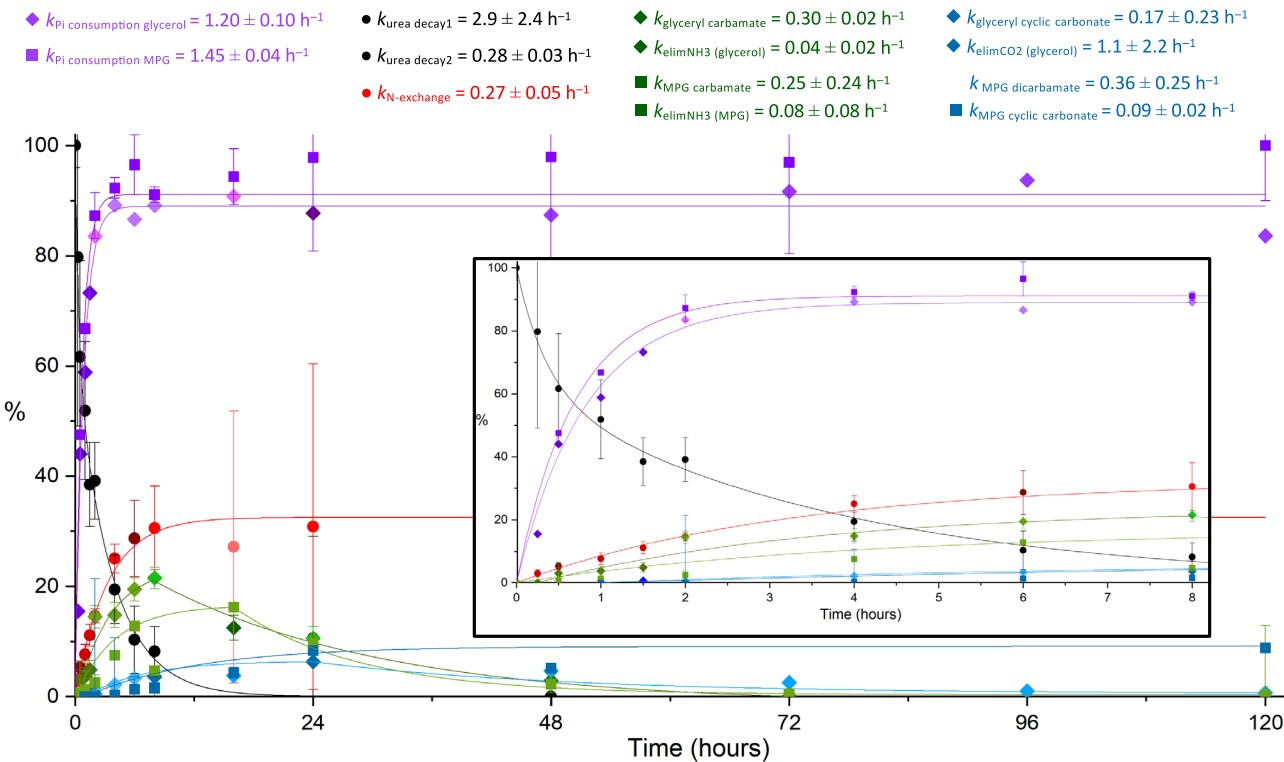

**Fig. 3 | Time dependence of $P_i$ consumption (cyan-violet), urea decay (black) and nitrogen exchange (red), amounts of glyceryl/MPG carbamate (green) and cyclic carbonate (blue) during phosphorylation of glycerol/MPG heated neat at 115 °C for 120 h [insert: 0–8 h].** Molar ratios either [$^{15}N_2$]urea/natural urea/ $NaH_2PO_4$/glycerol 0.5:0.5:1:1, natural urea/$NaH_2PO_4$/MPG or [$^{13}C$]urea/$NaH_2PO_4$/ MPG 1:1:1. Relative values [%] obtained from time-dependent NMR peak area integration (Supplementary Figs. 42–53), the error margins (bars shown) resulted from NMR signal-to-noise ratios (Supplementary Tables 9, 11, 12, 18–20). SNR-weighted exponential data fitting gave the associated apparent kinetic rate constants $k$ [per hour] (and half-times, see Supplementary Tables 15 and 23, and section 5.4 in the Supplementary Information). Different colour shades of symbols of the same fitted curve represent independent series of experiments.

extant biological membranes. Both, cyclisations and the formation of phosphodiesters require another phosphate activation and the elimination of a second water molecule. Very small amounts of $PP_i$ could be sometimes detected but hardly any sign of inorganic polyphosphates $P_{ni}$[46], no cyclic metaphosphates or branched ultraphosphates[47]. In six experiments at 3, 6 and 30 mmol scales (Supplementary Table 16), where equimolar mixtures of urea, $P_i$ and glycerol (**5**) were kept at 115 °C, 88–98% $P_i$ consumption were measured.

MPG phosphorylation at the same scales, viz. in 14 experiments at 3–10 mmol reaction scale, starting from equimolar neat mixtures containing natural urea (**2a**) or [$^{13}C$]urea (**2c**)/$P_i$/**6** (1:1:1), was sometimes less complete: 69–95% $P_i$ consumption were seen after 5 days at 115 °C (Supplementary Table 24). More than half of the MPG appeared depalmitoylated in these mixtures (Supplementary Figs. 30–32) but, on the other hand, dipalmitoylglyceryl phosphates were also found (Supplementary Fig. 33). Comparable amounts of carbamoylated **6** (when compared to carbamoylated **5**) appeared somewhat delayed (up to 16% after 16 h) and also cyclic MPG carbonate (about 9% after 24 h, Supplementary Fig. 53). At the 0.5 mmol scale we detected that $P_i$ consumption reached an end after 4 h showing an exponential rate constant at 115 °C that was essentially the same as in the glycerol phosphorylation reaction (Fig. 3). We compared this time-dependent $P_i$ consumption from $^{31}P$ NMR peak integration with the time-dependent decay of **6** from HPLC peak areas of the same crude mixture but extracted with methanol and calibrated using pure **6**. The exponential fitting on MPG consumption showed a thirteen-fold slower apparent rate constant than that of the corresponding $P_i$ consumption (Supplementary Table 23), which suggested the presence of some intermediate inorganic compound that could have accumulated

in the first 2–8 h of heating (Supplementary Fig. 53B). Nevertheless, owing to the necessary extraction procedure, we have never found direct spectroscopic evidence for the unstable inorganic reaction intermediates [$^{13}C$]cyanate (N$^{13}CO^-$) or [$^{13}C$]carbamoyl phosphate ($CP_i$), cf. section 4.4 in the Supplementary Information (SI). Notwithstanding, the transient presence of cyanate and $CP_i$ in these neat blends has been proven by the exchange of nitrogen isotopes on urea while heating the mixture in the actual reaction conditions. When starting with a 1:1 mixture of **2a** and **2b**, visible as a singlet and an overlayed triplet in the $^{13}C$ NMR spectrum, the appearance of a $^{13}C$-doublet urea resonance owing to the one-bond coupling with only one $^{15}N$ atom (Supplementary Tables 9 and 18) was observed only in the presence of $P_i$ (Supplementary Figs. 40 and 41 and Supplementary Table 8).

Scope of neat glycerol (**5**) and MPG (**6**) phosphorylation comparing different dehydrating conditions and inorganic phosphate sources (summary from sections 7.1–7.4 in the SI): We summarise here the most important features shown in the Supplementary Fig. 83 (**5** + $P_i$), Fig. 87 (**6** + $P_i$ at 115 °C), Fig. 88 (**6** + $P_i$ at 75 °C), Fig. 94 (**5** + cTMP), Fig. 97 (**6** + cTMP), Fig. 103 (**5** + $SP_i$ at 115 °C), Fig. 104 (**5** + $SP_i$ at 75 °C) and Fig. 111 (**5**/**6** + minerals containing $P_i$ or $PP_i$). **6** actually required the presence of **2a** to become efficiently phosphorylated at 115 °C, without urea at most 10% $P_i$ was converted to organophosphates in our neat, hot conditions. By contrast, **5** was phosphorylated even in the absence of **2a** showing $P_i$ conversions 36–77% after 4–5 days at 115 °C, presumably through the intermediate formation of $PP_i$. When $P_i$ and **5** or **6** were heated to a mere 75 °C for 5 days, 1:1 neat and in the absence of **2a**, zero $P_i$ consumption was seen with **6** and at most 2–4% $P_i$ consumption with **5**. In the equimolar

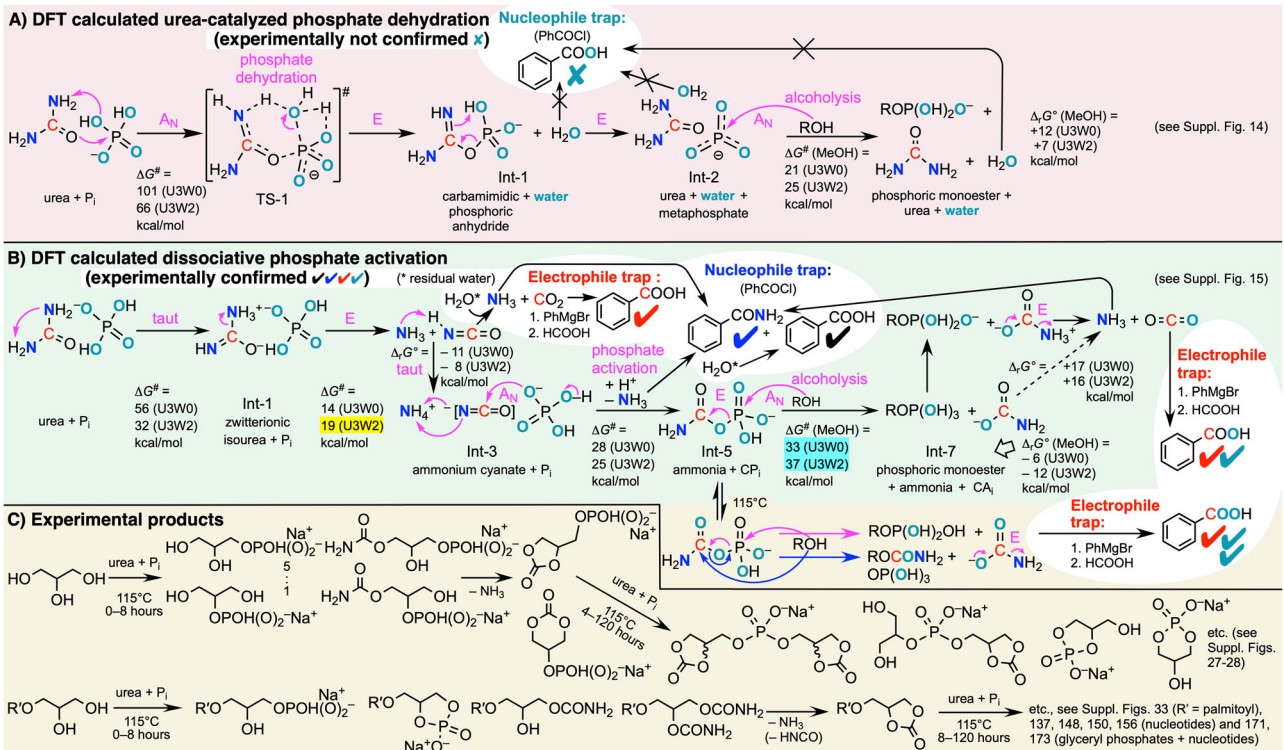

**Fig. 4 | Chemical reaction mechanism of urea-assisted phosphorylations using density functional theory and identified experimental products enriched with [13]C (red), [15]N (dark blue) and [18]O isotopes (turquoise). A, B** Representation of ab initio-computed structures (on pink and pale green backgrounds) showing the reactivity of urea + $P_i$ with methanol (MeOH) in the presence of two assisting urea molecules (total urea and water molecules = U3W0, assisting urea not shown) and, additionally, two assisting water molecules (= U3W2, assisting molecules not shown). Elementary steps (magenta): tautomerisation (taut), nucleophilic addition ($A_N$) and elimination (E). Most important calculated transition state (TS), intermediate states (Int) and associated free enthalpies of activation $\Delta G^{\#}$, and standard free enthalpies of reaction $\Delta_r G° = 0$ kcal/mol for urea + $P_i$ (U3W0) and (U3W2);

[kcal/mol] values from M06-2X/6-311++G(2d,2p)//M06-2X/6-31G(d,p) level of theory. Yellow text highlight = trajectory-determining transition state, i.e., fastest computed route (compare also with $\Delta G^{\#} = 24.5$ kcal/mol for the addition of a water molecule to the C-atom of N-protonated urea[41]). Cyan text highlight = $\Delta G^{\#}$ for methanolysis of carbamoyl phosphate $CP_i$ to methyl phosphoric acid and inorganic carbamate $CA_i$. **C** Time-dependent formation of main products (non-volatiles on pale ochre background, white zones for trapped volatiles) observed for the neat phosphorylation of glycerol (**5**) and MPG (**6**, R' = palmitoyl C16:0) using urea + $P_i$, isotope-labelled urea (**2b, 2c**) and [[18]O4]$P_i$. All details, including nucleotide products (structures not shown here), see Supplementary Information.

presence of **1** or **2a**, however, both **5** and **6** contributed to 65–77% $P_i$ consumption after 5 days at 75 °C.

This meant that evaporative conditions in arid environments with relatively strong temperature variations below and above the boiling point of water could lead to very efficient phosphorylations of both glycerol (**5**) and MPG (**6**) when urea (**2a**) was present. Such conditions may have been dominant on the first volcanic islands in the late Hadean global ocean, especially, under the influence of more intense and more rapid temperature variations during the short day-night rythm of that era.

With increasing molar excesses of urea, **6**/$P_i$/**2a** = 1:1:1; 1:1:2; 1:1:4; 1:1:10, we observed an increase in $P_i$ consumption up to 95% at 115 °C. Higher urea excesses expectedly led to more 5- and 6-membered ring cyclic phosphates. When **2a** was replaced with cyanamide (**1**), or $P_i$ was replaced with cTMP, **5** and **6** were efficiently phosphorylated at both 115 °C and 75 °C in the dry state. With cTMP at 115 °C, but not 75 °C, **5** became partly carbonylated even in the absence of **2a** or **1** (minimum 8.6%, cf. Supplementary Table 36). This meant that $CO_2$ from the air could be activated by cTMP at 115 °C and transferred to **5**, its glyceryl phosphate derivatives or **6**, to appear as 1,2-cyclic carbonates (cf. Supplementary Fig. 95). However, when **1**, **2a** or **2c** were replaced with equimolar amounts of one of the carboxamides, either formamide (**3a**), acetamide (**4a**) or their N-methyl substituted derivatives (**3c, 4c**, Fig. 1), the phosphorylation of **6** (measured as $P_i$ consumption) dropped to less than 10% unless two to fourfold molar excesses of **3a** were used at 115 °C, whereas the phosphorylation

efficiency of **5** by $P_i$ could be maintained at 60–82% in equimolar dry mixtures containing any of the carboxamides, but only at 115 °C. The use of 2 mol equivalents [[15]N]formamide (**3b**) or [[15]N]acetamide (**4b**) produced [15]NH3 at 115 °C that was trapped by benzoyl chloride.

Altogether, these experiments showed that the carboxamides did not accelerate the phosphorylation of **5** or **6** nor make the reaction more efficient. Instead, when **3a** partly degraded at high temperatures[43] and there was still more liquid formamide left to keep the "dry" solid-liquid mixture reasonably mobile, as was most likely the case when 2–4 mol equivalents **3a** were initially added—then only **5**, not **6** (nor any other alcohol), could be phosphorylated to a significant degree. When one of three minerals (or their analogue, cf. Fig. 1) were used as the phosphate source, that is, struvite (ammonium-magnesium $P_i$), vivianite (ferrous $P_i$) and the canaphite analogue calcium $PP_i$, we measured up to 74% $P_i$ consumption in the presence of **5** (Fig. 2 and Supplementary Fig. 111), which did not come by surprise, except that our attempts to phosphorylate **6** were unsuccessful, and the spectroscopic results from using natural vivianite powder ($Fe_3(P_i)_2$) should be taken with caution, both for analytical reasons.

We also compared mono-, di-, and trisodium phosphates ($P_i$) showing experimental pH values of equimolar mixtures with **5** and **2a** in water of 4.3, 8.9 and 12.8, respectively (Supplementary Table 35). The total conversion of $P_i$ into glyceryl phosphates in the absence and presence of **2a** grew from 0% (pH 12.8) through 36% (pH 8.9) up to 92% (pH 4.3) in accordance with the increase in protonation degree, i.e., tribasic < dibasic < monobasic $P_i$. By contrast, when compared to

urea-assisted reactions containing $NaH_2PO_4$ as the only phosphate source, tribasic sodium thiophosphate ($SP_i$ at pH 11.6) led to the phosphorylation of **5** under "dry" conditions when heated to 115 °C for 5 days, both, in the absence of any nitrogenous compound (22% glyceryl phosphates) and in the presence of equimolar **2a** or **1** (28–30% glyceryl phosphates) along with higher amounts of condensed inorganic and organic phosphates (9–35% $PP_i$ and 1% glyceryl diphosphates). The phosphorylation selectivity of primary versus secondary hydroxyls was between 3:1 and 4:1 using $SP_i$ (Supplementary Tables 38 and 39), thus, diminished when compared to $P_i$ or cTMP ($\approx$ 5:1). A mixture of mono- and tribasic inorganic (thio)phosphates in "dry" **5/2a**/$NaH_2PO_4$/$Na_3SPO_3$ = 1:1:0.5:0.5 (pH 7.5 in water) provided protons for both leaving groups, $H_2S$ from $SP_i$ and $NH_3$ from urea, which increased the total conversion to 36% at 115 °C, while the efficiency of an equimolar di- and tribasic dry mixture of **5/2a**/$Na_2HPO_4$/$Na_3SPO_3$ = 1:1:0.5:0.5 (pH 10.6 in water) remained the same as with $SP_i$ only (22% glyceryl phosphates) (Supplementary Fig. 103). The phosphorylation power of $SP_i$ remained substantial at 75 °C (Supplementary Fig. 104). When we added water right from the start, the conversion to phosphate esters dropped in most such "wet and evaporating" glycerol (**5**) mixtures to roughly one half after 5 days, when compared to those heated to 115 °C for the same period of time, but the phosphorylation of "wet and evaporating" aqueous MPG in **6/2a**/$NaH_2PO_4$/$Na_3SPO_3$ = 1:1:0.5:0.5 for 5 days at 75 °C remained beneficial (Supplementary Table 41). When a partly labelled equimolar neat mixture of $NaH_2P[^{18}O_4]$ and $Na_3SPO_3$, being approximately 50% $^{18}O$-enriched as in (**5** or **6**)/**2a**/$NaH_2P[^{18}O_4]$/$Na_3SPO_3$ 1:1:0.5:0.5, was kept at 115 °C for 5 days, we detected a diminished 31–39% abundance of $^{18}O$ isotopes in the major glyceryl and MPG phosphates (Supplementary Tables 40 and 42). This showed that, in a competitive situation without particular irradiation[19], but when sufficiently protonated, $SP_i$ kinetically outperformed the **2a** + $P_i$ couple by some 25–40% under the tested neat reaction conditions.

Neat nucleoside phosphorylation in the absence and presence of glycerol and alkanoic or amino acids (summary from sections 7.5 and 7.6 in the Supplementary Information): The results obtained from all our experiments on nucleoside phosphorylation in neat conditions in a mixed solid-liquid phase (Fig. 2), where only very small amounts of formamide (**3a**) have been added as liquidiser (we use this word to distinguish physically mobilising from dissolving), are comparable to previous works carried out in aqueous or formamide solutions. The relative proportions and amounts of phosphorylated ribonucleosides obtained at 115 °C are similar to those obtained by others from reaction mixtures heated up to 85 °C for several days in a urea-ammonium formate-water eutectic, liquid sulfur dioxide or subjected to dry-wet thermal cycling using large excesses of both urea and $P_i$ (10–16 mol equivalents), also more reactive phosphate sources richer in protons from ammonium ions[16,48], or drying down at 80 °C a hot paste containing amidophosphite and amidophosphates[32], and those in the presence of minerals, catalytic metal-doped clays, photoredox-active cerium phosphate, organo-catalytic 2,5-dimethyl imidazolidine-4-thione or in an initially acidic supercritical carbon dioxide-water ($scCO_2$-$H_2O$) two-phase environment[49–54]. In contrast to the nucleoside mixtures containing catalytic imidazolidine-4-thiones as in-situ phosphorylating agent[52], or those with 20 mol equivalents urea and 3 mol equivalents $NaH_2PO_4$ or hydroxyapatite heated at 94 °C in $scCO_2$-$H_2O$, where 5'-nucleotides dominated followed by carbamoyl nucleosides and acyclic 3'- and 2'-monophosphates[54], our hot neat solid-liquid conditions always produced a high fraction of 2',3'-cyclic phosphates (Supplementary Fig. 157). This fraction was also expectedly higher than that found in the presence of borate[55]. In an equimolar dry mixture, where all nucleosides were competing for limited amounts of inorganic phosphate but had enough urea at their disposal (U/C/G/A/$P_i$/urea/formamide 0.25:0.25: 0.25:0.25:1:4:0.5), about 75% of all nucleosides were converted to nucleotides after 5 days at 115 °C, out

of which almost 50% were cyclic and acyclic monophosphate esters of the nucleosides, i.e., 38.5% cNMP (N>p) and 9.5% 5'- and 2'/3'-NMP (pN + Np). About 25% were additionally phosphorylated and in varying parts carbamoylated cyclic and acyclic nucleoside phosphates (4.6% pN>p, 2.2% pNp), dinucleotide phosphodiesters (11% NpN + NpN>p + pNpN) and otherwise chemically modified products (8%).

Not much irreversible chemical nucleoside modification (denoted "other" in Fig. 2) occurred when heated neat at 115 °C for 5 days (Supplementary Figs. 148B and 150): hypoxanthine, inosine and inosylate were found by MS indicating adenosine $N^6$-hydrolysis, once xanthosine was identified, but not quantified, hinting at the odd guanosine $N^2$-hydrolysis (Supplementary Figs. 156–159). The proportions between pyrimidine and purine nucleotides, as deduced from a $UV_{260 nm}$-quantified HPLC-HRMS analysis, were little biased, but the composition showed a 4–5% hydrolytic preference for uridylates over cytidylates. In addition to this C-to-U hydrolysis, some inosine (I) and inosylate (IMP) hydrolysed from A and/or AMP, a little U dehydrated to $O^2$,2'-cyclouridine (cU) and inevitable depurination products (adenine, guanine, hypoxanthine), carbamoyl derivatives of the nucleotides as well as $N$-carbamoyladenine and $N$-carbamoylguanine were found, all of which produced the following rough molar ratios for the "canonical" (whether carbamoylated or not) nucleotides from initially strictly equimolar ribonucleoside mixtures: cytidylates/guanylates/adenylates/uridylates/others (purine bases, inosines, cU) = 20 : 23 : 26 : 29 : 2.

The phosphorylation of neat mixtures of nucleosides and their complementary nucleoside-5'-monophosphates (in the sense of Watson-Crick base pairing) did have an enhancing effect when compared to that of pristine nucleosides, especially in the presence of a purine nucleoside-5'-monophosphate. However, this enhancement was comparable to the efficacy of the phosphorylation and phosphodiester formation of an equimolar mixture of all four canonical ribonucleosides in the initial absence of nucleoside-5'-monophosphates (Supplementary Fig. 151). Dinucleotide (phosphodiester bond) formation represented typically between 2.4% up to 18.4% of the $P_i$ consumption (= conversion, cf. Supplementary Tables 47, 49, 50 and 51). No evidence for the formation of longer than dinucleoside phosphates has been found, neither by NMR spectroscopic (Supplementary Figs. 154, 165, 168) nor mass spectrometric means (Supplementary Figs. 148, 150, 156 and 171).

Of note, there were no proton-decoupled $^{31}P\{^1H\}$ NMR spectroscopic signatures of either poly-, cyclic meta- or branched ultraphosphates created in dry conditions[47], and no condensed nucleoside di- and triphosphates[56] could be found in the crude mixture containing all four nucleotides (no $\delta_P$-doublets and -triplets for NDP = ppN or NTP = pppN). Lowering the reaction temperature to 75 °C, replacing urea (**2a**) with cyanamide (**1**), or $P_i$ with cTMP or $SP_i$ brought nucleoside phosphorylation virtually to a halt, although pyrimidine nucleotides formed from **1** and $P_i$ to some notable extent (Supplementary Fig. 130). As shown by others, the phosphorylation of nucleosides with cTMP is comparingly weak, and requires dry-wet or hot-cold cycling in the presence of water, where sufficient magnesium or transition metal ions are mandatory[57,58].

All the above is consistent with previous works on the long-term hydrolytic stability of nucleosides[59,60] and sets a firm prebiotic rostrum for the early production of mono- and dinucleotide-type catalysts and scaffolds that could serve as feedstock for their refinement into the extant mono- and dinucleotidic enzyme co-factors, and also longer nucleic acids at a later stage of molecular evolution[61–63]. Our findings, that a significant amount of nucleoside phosphates was found carbamoylated when phosphorylated in urea + $P_i$ blends (Supplementary Figs. 137, 148, 150 and 156), especially adenosine at its exocyclic amino group, lends support to the thesis that a primeval translation could indeed have started with this kind of tether on $N^6$-carbamoyl adenylate-bearing oligoribonucleotides[64].

Remarkably, the presence of L-valine suppressed neat D-ribonucleoside phosphorylation by about 34%, whereas the presence of

D-valine suppressed it by merely about 11% (Supplementary Fig. 162 and Table 55), which could be explained by the fact that L-amino acids more tightly interact with D-ribonucleosides than D-amino acids do. This is consistent with the findings that naturally configured oligoribonucleotides direct the synthesis of naturally configured peptides more readily than of their enantiomeric counterparts[65,66] and L-amino acids tend to generate a bias in favor of D-glyceraldehyde from formaldehyde and glycolaldehyde[67].

Neat hot equimolar mixtures of all ribonucleosides, glycerol and four alkanoic (fatty) acids in one pot produced within 48 h up to diglyceryl, dinucleoside and glycerylnucleoside phosphates (cf. section 7.6 in the SI, Supplementary Figs. 163–174). Striking in this experiment was the fact that we could not find any sign of carboxyester formation, that is, no sign of glyceryl esters of any of the prebiotically plausible alkanoic acids. This makes us conclude that other prebiotic sources, ammonia-poor reaction conditions or environments, must be searched for the making of mono-alkanoic glyceryl esters (such as MPG) from glycerol, or glyceryl phosphates, and alkanoic acids, also other sources than from the partial deacylation of diacylglyceryl phospholipids such as BTG or DOG phosphates, that gave in only very small amounts 3-tridecanoylglyceryl-1,2-cyclic phosphate (5cTGP) and 3-monooleoylglyceryl-1,2-cyclic phosphate (5cMOGP), see below.

Phosphorylation of the monools 7, 8, 9, 10 and pyruvic acid 16 (summary from sections 7.7–7.11 in the SI): We chose the monofunctional model alcohols 7–10, two of which are diacylglycerols (7, 8) and two alk(en)yl alcohols (9, 10) for two reasons, prebiotic and mechanistic. BTG (7) is a plausibly prebiotic precursor because of its relatively short and fully saturated long-carbon-chain residues, with a high membranogenic potential in the context of a phospholipid; but 7 is for the same reason quite an immobile material, which may have had an important bearing on the neat reaction conditions that we studied. Therefore, we chose to test in addition a chemically similar yet much less prebiotic model alcohol, DOG (8), unlikely to have existed in the late Hadean because of its *cis*-configured unsaturations (*trans*-alkenes are more stable). Its oily aspect, however, would allow us to conclude on whether the higher physical mobility, or rather the chemical reactivity of a single alcohol group would impact more on the outcome of the tested neat, hot phosphorylation reaction conditions. Alkanols such as dodecan-1-ol (9) could have been precursors of long straight-chain alkyl phospholipids without glycerol in between the chain and the polar headgroup, that could assemble to primitive versions of bilayer membranes[68]. An alternative version of primeval membranogenic phospholipids may have been Archaea-like branched isoprenoid phosphates the prebiotic precursors and membranophilicity of which just have started to be explored[19,69]. Both kinds of alcohols, unbranched and branched, are primary alcohols. Dodecanol (9) and geraniol (10) are monools that therefore served us as model precursor compounds of phospholipids for bilayer membranes involved in the evolution of cell walls of Bacteria and, respectively, Archaea at some later stage. Pyruvic acid (16) is a recognised crucial protometabolite that could have been generated under prebiotic conditions[70].

A sobering fact was that the monools 7, 8 and 10 were not phosphorylated in dry (neat) equimolar mixtures of P$_i$ and urea after heating at 115 °C, if not for organophosphates in barely detectable trace amounts mostly of deacylated 7 (Supplementary Fig. 178) and 8 (Supplementary Figs. 182 and 183). Previous work did show that significant amounts of 8 could be phosphorylated under conditions akin to those studied here, but only using a large excess urea (2a) with respect to the starting alcohol 8, combined with an excess of more proton-rich phosphate sources: ammonium dihydrogenphosphate or phosphoryl ethanolamine[71]. Heating 9 and aqueous P$_i$ in the presence of up to 10 equivalents 2ac at 115 °C for 4−5 days generated no organic phosphates whatsoever, only inorganic condensed phosphates PP$_i$ and PPP$_i$. By contrast, when neat 9/1/P$_i$ (1:1:1) mixtures were heated at 115 °C, the P$_i$ consumption was almost 80%, some organic phosphates

(Supplementary Fig. 203) and quite persistent carbamoylated products were found, even more of the latter at 130 °C (Supplementary Fig. 191). In the labelled mixtures using 9/1/[$^{18}O_4$]P$_i$ we have found $^{18}O$-enrichment in the not phosphorylated dodecyl carbamate and carbonate side-products as well (Supplementary Tables 61 and 62). Evidently, the intermediate inorganic carbamoyl phosphate (CP$_i$) slowly exchanged $^{18}O$ with $^{16}O$ isotopes prior to dodecanol attacking the carbon or phosphorous atoms of CP$_i$ (cf. ochre zone of Fig. 4B). This attack on the carbon atom of CP$_i$ did not happen to any of the diols and triols 5, 6 and 11–15, since no $^{18}O$ enrichment in the corresponding not phosphorylated side-products was found whatsoever. This, in turn, means that the alcoholysis of CP$_i$ by means of an attack of an OH group of a diol or triol at the phosphorous atom of CP$_i$ is just too fast, accelerated by the neighbouring OH group, to allow for a competitive attack on the carbonyl group of CP$_i$. When 1 and P$_i$ were used on neat 10, it generated what appeared to be 5-membered ring cyclic phosphates, indicating the addition of P$_i$ to geraniol's first C=C double-bond, and other less well characterised by-products; probably cyclic monoterpene and geranyl-cyanamide derivatives (Supplementary Fig. 209B). Finally, in hot neat mixtures containing 16 the inorganic phosphate sources P$_i$, cTMP and SP$_i$ condensed separately to PP$_i$, PPP$_i$ and, respectively, SPP$_i$, SPSP$_i$ and SPPP$_i$ (Supplementary Figs. 211, 213 and 214) while 16 oligomerised to create, through up to four new C–C bonds, novel polyketides (Supplementary Fig. 222), some of which appeared to be partly phosphorylated (Supplementary Fig. 215).

For the geochemical context: by far the easiest, environmentally most robust and atom-friendly way of phosphorylating polyfunctional alcohols such 1,2,3-triols, 1,2-diols and 1,3-diols like 5, 6, 11–15, but not monools like 7–10 and 16a, accompanied by a minimal quantity of wasted atoms under conditions that involve hardly any water, no molar excess of inorganic phosphate over the alcohol, no excess ammonia, imidazole[15] or liquid sulfur dioxide[50] and no direct intervention of transition metals, borate minerals, silica or clays, or super-critical carbon dioxide[51,53–55], is the presence of large amounts of hot urea (2a). Abiotic urea consists of a thermally stabilised solid form of reactive ammonium cyanate that, by irreversibly evaporating ammonia and leaving cyanate to react, actually drives the phosphorylation reaction at temperatures close to that of boiling water and above (and below). Even at a mere equimolar ratio of alcohol and phosphate, urea can act as an effectual chemical fuel of phosphorylation that will "burn" anyway, that is, hydrolyse in the presence of water to produce ammonia and carbon dioxide. Therefore, out of the here tested dehydration and phosphorylation agents, urea in combination with P$_i$ (= 2a + P$_i$) are geochemically the likeliest environment for prebiotic phosphorylations of simple organic precursor compounds, precisely because they bear the lowest chemical potential of all: solid urea-P$_i$ mixtures could have acted on organic molecules for the longest periods of time before vanishing through hydrolysis. The urea-assisted phosphorylation of glycerol (5), MPG (6) and the nucleosides (11–15) produced carbamoyl- and cyclic carbonate derivatives in addition to the phosphate esters; those carbonates generated from 1,2-diols persisted in 5–15 mol% fractions even 5 days at 115 °C in the dry state.

The tracking of $^{18}O$ isotopes showed that there is no organo-catalytic potential in urea (Fig. 4A) when it comes to phosphorylations, the only operative reaction pathway is the thermal dissociation of urea to ammonia and carbamoyl phosphate (Fig. 4B and section 6.3 in the SI). Our ab initio computations have shown that urea's interactions with water and methanol are relatively weak when compared to those with inorganic dihydrogenphosphate (P$_i$). The dissociative phosphate activation pathway (Fig. 4B) clarifies, how P$_i$ accelerates as a so-called acid-base catalyst the degradation of urea to carbon dioxide and ammonia, viz. through catalysing the direct or water-relayed formation of a zwitterionic tautomer of urea (→ Int-1 in Fig. 4B) predisposed to break one carbon-nitrogen bond and generate ammonium cyanate NH$_4$NCO (TS_2 and Int-2 in Supplementary Fig. 15). The computed

potential energy surfaces give a rationale for how inorganic carbamoyl phosphate ($CP_i$) is generated in neat urea-phosphate **2a** + $P_i$ mixtures through transitional $NH_4^+ + NCO^- + P_i$ clusters (Int-3 in Fig. 4B and Supplementary Fig. 15), and how $CP_i$ acts as a universal reaction intermediate with alcohols (Int-5). The rate-limiting free enthalpy of activation for the addition of a water molecule to urea protonated on one N-atom by $H_3O^+$ in acidic water, which was recently shown to then break urea's C-N bond and thermodynamically unfavorably lead to ammonia ($NH_3$) and inorganic carbamic acid ($CA_i$) in the absence of $P_i$, was calculated to be higher when computed at the MP2/aug-cc-pVTZ level of theory[41]. Unlike phosphorylation reactions taking place in water[11], those in neat urea as well as in a milieu containing both urea and water (3:2 mol/mol) are favorable and calculated to be exergonic in the beginning formation of organic phosphoric monoesters, even though the decomposition of $CA_i$ to $NH_3$ and $CO_2$ was calculated to be endergonic by 16–17 kcal/mol (Int-7 → Int-8 → methylphosphate + 2 $NH_3$ + $CO_2$). Of course, it happens to completion anyway in an open system like the Earth's surface. Hence, both kinetically and thermodynamically, the pathway in which urea dissociates into ammonium cyanate wins (Fig. 4A versus 4B and Supplementary Figs. 14 vs. 15). However, the alcoholysis of $CP_i$ was calculated to be quite difficult for methanol (highlighted in cyan in Fig. 4B), which is consistent with our experimental outcome that 1,2,3-triols, 1,2-diols and 1,3-diols like **5, 6, 11–15** are incomparingly more efficiently phosphorylated than the monools **7, 8, 9, 10** and enol **16a**.

Cyanamide (**1**), too, can directly react with inorganic phosphate and alcohols to produce phosphate esters and carbamates (Fig. 2 and Supplementary Figs. 83, 86–88, 97, 103, 104, 111, 130, 131, 181, 184, 193, 194), but can also add to alcohols without the inference of phosphate, cf. section 7.11 in the SI (also Supplementary Figs. 200–202, 207–210, 216 and Supplementary Table 61). However, monobasic (slightly acidic) phosphate is known for long to efficiently catalyse the hydrolysis of **1** to **2a**[40]. Therefore, especially during cool night-time and water-condensing or rain periods, cyanamide (**1**) should be seen for kinetic reasons to primarily take the geochemical role of being an important precursor of urea refurbishment. The tested formamides (**3abc**) and acetamides (**4abc**) are no alternative as urea surrogates, but formamide can enhance, as a substitute for liquid water, during very hot day-time conditions, urea-assisted phosphorylations of the only tested alcohols that remain solid (immobile) at 70–115 °C, viz. the nucleosides. Chemically internally activated inorganic phosphate sources, such as condensed phosphates $P_{ni}$ ($n = 2, 3, …$) generated by heat (trimetaphosphate, cTMP) or crystallised in minerals (canaphite), and thiophosphate ($SP_i$) present in hydrogen sulfide-containing acidic volcanic waters during hours, days or longer[19], can indeed phosphorylate alcohols, but not as long-lasting as urea and inorganic orthophosphate (**2a** + $P_i$). It seems that, over longer episodes including aqueous wet periods, the condensed $P_{ni}$ were mainly hydrolytic reservoirs of $P_i$. In an alkaline milieu, however, cTMP is a potent and valid phosphate source, particularly, in contrast to the **2a** + $P_i$ couple or $SP_i$ that actually profit from available protons and slightly acidic conditions. Only cyanamide (**1**) has a higher chemical potential than cTMP (Fig. 1); cTMP at 115 °C is able to form cyclic carbonates from atmospheric $CO_2$ and 1,2-diols present in diglyceryl phosphates (Supplementary Figs. 66–69). Carbonylisation stabilises such organic phosphate esters and, as mentioned before in the context of hot urea + $P_i$ blends, makes them more resistant against degradation over longer periods of time. However, ribonucleosides and monools are hardly or not at all phosphorylated by cTMP in alkaline neat conditions, at least without abundant magnesium or transition metal ions[57,58]. Both, $SP_i$ and cTMP degrade to $P_i$ when the pH drops, which is a key argument in the context of an atmosphere that bore huge amounts of carbon dioxide that would importantly acidify the waters by hydrolysing to aqueous carbonic acid[72]. Therefore, the best immediate prebiotic value of cTMP is its role as a persistent

condensing agent based on its efficiency in creating peptides from amino acids[46,73,74], and that of $SP_i$ could be drawn in particularly alkaline arid environments, where the urea + tribasic $P_i$ blend is an unreactive phosphorylation agent while tribasic $SP_i$ still is (section 7.4.1 and Supplementary Figs. 74–78), and from its photochemical reducing power[19].

In summary, this work shows that, in conditions that were poor in available (soluble) phosphate sources and limited in non-volatile reduced-nitrogen compounds—as should be expected to have been the case on the earliest subaerial landmasses being scarce in the late Hadean era—single-chain (mono-alkanoyl) glyceryl phosphate derivatives appear to be the most likely chemical starting line for the first robust bilayer-forming phospholipids. As a preliminary result to be followed more systematically, we have shown by epifluorescence microscopy that crude extracted, dried and rehydrated phosphorylation mixtures, where MPG, $P_i$ and urea (1:1:1) have been heated to 115 °C for 5 days in the dry (neat) state, did form giant vesicles showing diameters of 10–30 μm (Section 8 in the SI). Hence, when episodically wetted with salty water, mono-alkanoyl glyceryl phosphates could have let vesicles emerge that enclosed short nucleotides, amino acids and perhaps peptides[75–78] inside them. It is less clear at present exactly how alkanoic "fatty" glyceryl esters and longer than dinucleotides could have amassed in such arid conditions, perhaps through wet-dry cycling. Most if not all of the prebiotically available monool starting compounds, and their derivatives that would be generated, for instance, in neat hot inorganic phosphate-urea blends, could have taken some role as a co-surfactant in primitive membranes, stabilising or modulating their permeabilities, shapes (positive versus negative curvatures), membrane-membrane interactions (vesicle fusion versus cluster formation), growth and division (ease of fission) and robustness (chemical and physical)[68,71], but they are unlikely to have been the major players irrespective of whether they were glyceryl derivatives or not. Monools must have been outcompeted by prebiotically available mono-alkanoyl glycerols, glycerol, and probably also carbohydrates that weren't studied here. Taken together, we have constrained the geochemical space for the earliest possible prebiotic phosphorylation reactions, setting the stage for their assembly to form protocells in shallow basins once liquid water arrived more recurrently[79] and stayed longer in place, while increasingly notable concentrations of other dissolved small-molecular-weight precursor molecules, such as α-keto acids, α-amino acids, short peptides and carbohydrates, and N-heterocycles, were able to impact on the protocell's content[16,70,80–82].

Our results should provide valuable data for the theory on the outbreak of life in seafloor hydrothermal-sedimentary systems[83] or deeper hydrothermal-vent systems[84,85] that offer dielectric barriers as compartments for efficient chemiosmotic work to be casted[86] albeit, as such, with a limited evolvability potential. We should think of the possibility that probably neither of these initial subaerial and seafloor systems could build up sufficient complexity to become alive on their own. With time passing, the average subaerial surface temperature diminished and the impact and radiation intensities from outer space steadily calmed down[87], which makes it likely that the subaerial locations eventually began to lack sufficiently congregated long-term energy sources to unfailingly push forward the disequilibrium to a further level of complexity. Because of imperatives imposed by the thermodynamics in liquid water, the seafloor was not a locum apt to amass enough organic phosphates from the elimination of water molecules out of their precursor compounds. We need to consent to a common leeway in which the subaerial and the seafloor locations physically communicated with one another, given the Hadean weather, through strong winds[28] bearing microdroplets[41] and aerosols[88,89], and the intense rhythm of flood and ebb, back and forth to the ocean, of water that contained sinking phospholipid protocells originating from subaerial milieus, arid conditions and harbouring dense amounts of nucleotides and peptides. Once sunken to the ocean ground, they

could fuse with seafloor compartments[90,91] and take over their sheltered proto-metabolic reaction networks[20]. Only the combined ability to produce true polymers by installing the recurring translation of a simple genetic code from scarce but well soluble nucleic acids (polyphosphodiesters) into functional proteins (poly-carboxamides) more robust and more abundant in numbers, could provide and sustain more, and more specific catalysis from proteic, early enzymes. Thereafter, the first primary metabolic networks could detach from the seafloor, spread into the ocean, thus, become more independent of mineral catalysts and their primitive energy sources. And probably only these next-level entities could become truly self-evolvable over periods of time much longer than was the average lifetime of their molecular components[92].

## Methods

### Experimental materials

Pure racemic MPG (**6**) and racemic BTG (**7**) have been synthesised and spectroscopically characterised (cf. sections 1.6.1–1.6.3 in the SI). Racemic DOG (**8**) was available from our earlier work[71]. Solid sodium dihydrogenphosphate highly enriched in the $^{18}O$ isotope, $NaH_2P[^{18}O_4]$ ( $= [^{18}O_4]P_i$), was prepared from $PCl_5$, $H_2[^{18}O]$ and $Na_2CO_3$ according to a new protocol established ad hoc (section 1.6.4 in the SI). All other organic or inorganic starting compounds were commercially available.

### Experimental methods

Several hundred milligrammes to several grammes of the neat components (0.5 to 30 mmol per compound, cf. reaction scales in Supplementary Tables 16 and 24) were pre-mixed usually in equimolar ratios, sometimes dissolved in water, lyophilised or let evaporate to a visibly dry state, yet containing residual amounts of water (often about 15 w%, cf. Supplementary Table 25 and section 5.4 in ther SI). These neat mixtures were kept in an open system for up to five days at a constant 60, 75, most often 115 or, when using dodecanol also 130 °C (section 2.1 in the SI). Each reaction condition was carried out in triplicate. In those experiments where the reaction kinetics and stable isotope distribution was followed, the neat reactions were carried out in a closed system under a steady argon flow to carry over volatiles to two traps containing reactive solutions (toluene containing excess benzoyl chloride in trap 1, and tetrahydrofuran containing excess phenylmagnesium bromide in trap 2). After the heating period, the trap liquids for volatiles were periodically sampled, trap 2 liquid acidified with formic acid, and diluted in a standard solvent mixture for LC-MS analysis (section 2.2 in the SI). The non-volatiles present in the heated neat reaction mixtures were sampled by extraction with either dimethyl sulfoxide or DMSO-*d6*, water or $D_2O$, or in some cases methanol (cf. respective sections 1.2, 1.3, 7.6 and 8 in the SI) for their subsequent analysis: $^1H$, $^{13}C$ and $^{31}P$ NMR, 2D correlation including diffusion-ordered $^{31}P\{^1H\}$ spectroscopy (DOSY), and LC-MS using reversed phase and hydrophilic-ion liquid chromatography (HILIC) (ultra)high-performance separation techniques (HPLC and UHPLC) coupled with a UV-spectroscopic and an evaporative light-scattering detector (ELSD), or an electrospray-ionisation-high-resolution mass spectrometer (ESI-HRMS). The results of essentially all of the spectroscopic, chromatographic and spectrometric data of more than 430 individual experiments are recapitulated in Figs. 2–4C, and presented in a detailed and processed form in the sections 4–7 of the SI, accompanied by an appropriate supplementary text and conclusions. The preparation of lipid films from crude phosphorylation mixtures, of lipid vesicles using the natural swelling method, and fluorescence microscopy images thereof, are shown in section 8 of the SI.

### Theoretical methods

A quantum theoretical study using the Density Functional and Coupled Cluster Theories was carried out on an ab initio-assembly of dihydrogenphosphate (1 molecule), urea (3 molecules), water (0 or 2 molecules) and methanol (1 molecule). We performed a comparative analysis along the calculated Intrinsic Reaction Coordinate using the M06-2X and CCSD methods in combination with the 6–31G(d,p) and 6–311++G(2d,2p) basis sets. The results of four distinct reaction pathways, calculated at the M06-2X/6–311++G(2 d,2p)//M06-2X/6–31G(d,p) level of theory are presented in a highly condensed form in Fig. 4A, B. The whole computational study is detailed in section 1.5 (quantum theoretical method) and section 3 (detailed results) of the SI, accompanied by an appropriate supplementary text and conclusion. The Gibbs free energies of all computed stationary (transition and intermediate) states are listed in Supplementary Tables 4–7; the atomic coordinates optimised at the M06-2X/6–31G(d,p) level of theory are made available as a separate Supplementary Dataset.

## Data availability

All data needed to support the conclusions of this manuscript are included in the main text, the Supplementary Information and in 8 Supplementary Datasets.

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

## Acknowledgements

We thank Robert Pascal (*Université Aix-Marseille*) for his initiative and support in the earliest stages of our work on the reaction mechanism, Peter Goekjian (*Université Claude Bernard Lyon 1*) for insightful NMR interpretations, and Marc Vedrenne (*Institut de Chimie de Toulouse, Université Paul Sabatier*) for taking NMR spectra using a quadruple-resonance inverse-detection cryoprobe. The financial supports from the *Volkswagen-Stiftung* (Az 92 850 for P.S.) and the LABEX *Institut Lyonnais des Origines* within the program *Investissements d'Avenir* of the French government (LIO PhD program 2020 for A.S.), operated by the *Agence Nationale de la Recherche* (ANR-10-LABX-0066 for I.D.), are gratefully acknowledged.

## Author contributions

P.S. designed and supervised the research, and wrote the paper. M.F. and I.D. helped to design the study and with supervising A.S. H.C. supervised the ab-initio calculations and helped with writing the quantum theoretical sections. A.S. performed all experiments, analyses, curve fittings, designed Fig. 2 and wrote the Supplementary Information (SI) document, quantum theoretical and conclusion sections exempt. L.M. conceptualised and performed all ab-initio calculations and the reaction pathways depicted in the SI. A.L. analysed lipids for the reaction kinetics. E.F. performed and analysed with A.S. the mass spectra. A.B. performed and analysed with A.S. the NMR spectra. M.F. synthesised with P.S. labelled phosphate, with A.S. and A.L. reference compounds, performed the experiments on dodecanol, the microscopy of vesicles and contributed to writing the SI.

## Competing interests

The authors declare no competing interests.
