## [Transparent Peer Review file · Nature Communications]

A scope of prebiotic neat reaction conditions and the mechanism of urea-assisted phosphorylations of alcohols

Corresponding Author: Professor Peter Strazewski

Version 0:

Reviewer comments:

Reviewer #1

(Remarks to the Author)

This study definitively establishes how urea can assist the phosphorylation of alcohols and nucleosides. The work is comprehensive, employing a multitude of experimental techniques and a detailed SI that provides specific details. I was asked to provide feedback on the computational part as I am not an expert in experimental measurements. I have read the paper in its entirety and found the writing style challenging to follow. Many sentences attempt to convey too much information in a single location.

In terms of the computational aspect, the authors analysed three PES, which are detailed in Fig. S13, S14 and S15 of the SI. Fig. S13 shows the pathways for associative phosphate dehydration with and without extra water molecules. Fig. S14 and S15 show a dissociative phosphate activation path without/with added extra water. All these PES have been characterised at the DFT level using the M06-2X functional, which includes London dispersion interactions. This is important as the considered molecular aggregates are relatively large and dispersion may act as a compacting factor. A modest 6-31G(d,p) basis set was adopted for all calculations (geometries, energies and thermodynamic corrections).

1. The "energetic" must be refined in terms of both the level of calculations and the basis set. All the considered PES envisage a negatively charged system, for which diffuse functions are essential. A single point energy evaluation for all stationary points and TS at M06-2X/6-311++(2d,2p)//M06-2X/6-31G(d,p) is the best approach. I recommend running a single point energy evaluation at CCSD(T)/6-311++G(2d,2p)//M06-2X/6-31G(d,p) for the reagent, first and highest TS, and the corresponding intermediate of the most promising path (Fig. S15). This will allow us to ascertain whether M06-2X is suitable for this type of rearrangement. A potential correction to the M06-2X for the whole PES can then be made.

2. The captions for Figs. S13-15 miss the energies to be Gibbs free energy as also commented in the text.

3. The addition of methanol (the simplest alcohol to be phosphorylated) at a specific point of the PES is an important and striking addition. This is not achievable in practice as the alcohol is always present and can interfere or play a role similar to extra water in the early stages of the PES. The authors' proposal simplifies the whole PES, and I understand their reasoning. However, this merits a comment and some hypotheses of the potential changes due to its presence from the very beginning.

4. If I have understood correctly, the authors mimic a situation in which water is almost absent. This means the reactions are carried out in an almost solid-like environment where the molecular mobility is reduced. The DFT calculations instead leave full mobility during the geometry optimisation. Having forced some geometrical restraints will further increase the kinetic barrier. The authors must comment on this.

I will be happy to review again this section of the work in the revised version.

Reviewer #2

(Remarks to the Author)

The manuscript entitled 'A scope of prebiotic neat reaction conditions and the mechanism of urea-assisted phosphorylation of glycerols, other alcohols and nucleosides' summarizes the results from phosphorylation experiments with model alcohols under various reaction conditions.

Here are my comments:

In the first sentence of the manuscript the authors make already a quite bold statement: 'Two theories of the origin of life on Earth, one located in the Hadean seafloor, the other on the surface of subaerial landmasses and basins, need

reconciliation.' I do not agree with this. Firstly, I would not call different (reaction) conditions a theory and secondly, why do these spaces have to be brought together? I think it is only due to our limited imagination that we are not able to accept different conditions. It is interesting to see that many find it difficult to accept different reaction conditions. Even on our planet there are such large temperature differences, pressure differences, if we look at the oceans or volcanism. This needs to be rephrased.

Ultimately, the authors themselves recognize the dilemma: 'However, the lately growing number of different prebiotic dehydrating conditions proposed to have furnished crucial organic phosphate esters shortly before the first living cells have emerged in the late Hadean and early Archean eras, leave at present the scientific community with a very large range of possibilities that lack thus far appropriate experimental comparison.' Therein lies the mistake, you don't have to compare the conditions but investigate whether such reactions are possible under given conditions. There are very good studies in literature that do so, even if some researchers immediately criticize it if there is no 'high yield' reaction.

The following papers should be cited in the context of oxygenated precursors:

Beyazay, T. et al. Ambient temperature CO₂ fixation to pyruvate and subsequently to citramalate over iron and nickel nanoparticles. *Nature Communications* 14, 570 (2023).

Peters, S., Semenov, D. A., Hochleitner, R. & Trapp, O. Synthesis of prebiotic organics from CO₂ by catalysis with meteoritic and volcanic particles. *Sci. Rep.* 13, 6843 (2023).

Interestingly, the authors consider dissolving insoluble phosphates with amines or nitrogen compounds. Chemically speaking, this makes no sense. Why not acids? Moran has shown that acids are also formed, even complexing acids. They even dissolve apatite. This is known from lemonades containing citric acid, which damage the teeth.

With regard to the phosphorylation of amino acids and other compounds, it would be advisable to search the chemical literature and not just the geochemical literature (in part in French(!)):

Rabinowitz, J. Recherches sur la formation et la transformation des esters lxxxiii [1]. réactions de condensation et/ou de phosphorylation, en solution aqueuse, de divers composés organiques à fonctions OH, COOH, NH₂, ou . *Helv. Chim. Acta* 52, 2663-2671 (1969).

Rabinowitz, J. Note on the role of cyanides and polyphosphates in the formation of peptides in aqueous solutions of amino acids, at room temperature, as a possible prebiotic process. *Helv. Chim. Acta* 54, 1483-1485 (1971).

Hatanaka, H. & Egami, F. The formation of amino acids and related oligomers from formaldehyde and hydroxylamine in modified sea mediums related to prebiotic conditions. *BCSJ* 50, 1147-1156 (1977).

I also noticed that phosphorylation with small organic molecules is not mentioned at all. Considerable selectivities have been reported here recently. This applies in particular to imidazole derivatives, imidazolidine-4-thiones and 2-aminooxazole. With regard to the selection of alcohols, even more interesting representatives would come to mind later on, as lipids come rather late in the chemical chain.

Please correct the reaction scheme leading to benzoic acid in Fig. 4. A. It looks strange when benzoic acid is formed from other molecules out of the blue. The same holds in part for Fig. 4.B.

I would rename the title of Figure 4 and also in the text. It is only a mechanism in part (TS in Fig. 4. A), it is more a proposed reaction cascade.

If the authors really investigated the mechanism of the reaction experimentally, I would expect the reaction orders for the reagents as a minimum requirement. Does it agree with the proposed model derived from the DFT calculations?

DFT calculations are mentioned, also the cartesian coordinates in the SI. Please provide information about the software used and the basis set, functionals etc..

The diastereomeric interactions are indeed surprising.

Please check the references. I have noticed that certain references are missing in the text.

For me, there are too many open questions to make such conclusions from the experiments (Our results should provide valuable data for the theory on the outbreak of life in seafloor hydrothermal-sedimentary systems). I think that's completely exaggerated.

Because of these points, complete revision is necessary, and I do not consider the results to be as new as they are presented here.

Reviewer #3

(Remarks to the Author)

Phosphorylation of alcohols and nucleosides is one of the most interesting and challenging topics in prebiotic context. In this paper "A scope of prebiotic neat reaction conditions and the mechanism of urea-assisted phosphorylation of glycerols, other alcohols and nucleosides", the authors promoted the phosphorylation of different compounds using urea/cyanamide and other "auxiliaries" to achieve the phosphorylation of glycerol, other alcohols and nucleosides.

Overall, the results are interesting and present a very detailed study with a lot (a lot) of information (some of that information is not even mentioned in the main paper), nevertheless, I would recommend publishing this manuscript, but the presentation of the results can be improved if they clarify the reasoning of some experiments, some statements on the discussion and finally the final conclusion.

Major concerns:

- Reading the main paper is a difficult task, there is not any specific order, for instance in the results section immediately, they refer to figures S142, Fig. S195 (but the SI finished in Fig. S194). Then mentioned Figs. S157 and S160-S163, Tables S55 and S56. In my opinion the manuscript needs to be written in a sense that it would be “easy” for the reader, not only for the understanding but for the flow of information.
- Normally, for a SI file the authors should ensure that the information is completely clear and well presented, but it is difficult to follow. The sections are mixed, like first experiments with glycerol and MPG and urea, then change to glycerol alone then move with cTMP with both and finally with the others P sources. It would be great if they can write the sections based on the alcohol or nucleoside. It means, all experiments with glycerol, then with MPG and the rest of the alcohols (7, 8, 9 and 10), and finally with the nucleosides.
- Following the above comment, just the NMR of urea assisted phosphorylation is shown. I understand that all NMR will be a huge file, but at the end of each section, the authors (in just a figure) can stack the ^1H and the ^{31}P NMR, where shows the comparison of glycerol + Pi and the condensing agents, or glycerol + urea + the P sources, and so on.
- Basically, all the data of the rest of the reactions (other P sources, no any data of the liquidisers) are in tables of bar graphics, it would be interesting to see the NMR (as mentioned above).
- The major concern of the manuscript is the quantification for the conversion % (based on the main paper: “the phosphorylation yields of 6 (measured as Pi consumption)”. It is well known that ^{31}P qNMR is a valuable analytical technique for evaluating some phosphorus-containing compounds. But I did not find any standard (internal or external) that the authors used for the quantification.
- They said that for qNMR 60 s of delay was used, how many scans they used? Some figures of the ^{31}P NMR show a high signal:noise ratio. This inconvenience would interfere with the quantification.
- There are several references that use ^{31}P qNMR for quantification (see these two: Anal. Chem. 2024, 96, 11198–11204 and Appl. Sci. 2025, 15, 323), and as it can be seen the use of a standard is almost “a must”. If the authors did not use a standard, that implies that they can not claim “yields”, and if they just mentioned conversion % of Pi (or the others P sources) that is a mistake as well. Some explanations for these comments:
- Figure S16 shows the ^1H NMR spectrum (500 MHz of the DMSO- d_6 extract of a mixture of glycerol (5), natural isotope abundance-urea (2a) and NaH_2PO_4 (Pi) (1:1:1, 0.5 mmol each) after heating it neat for 120 h at 115 °C), these are the net conditions for most of the experiments. Immediately, it can be seen the amount of unreacted glycerol but based on the figure 2 (main paper) and the table S26 (same conditions) the total conversion of Pi to phosphorylated organic products (glycerol in this case) was 91%. If they try to use the deconvolution for the ^1H NMR of the reaction and obtain the conversion % of glycerol-to-glycerol phosphates the difference will be huge. I don't get it if the 91% of the Pi was transformed into the glycerol phosphates why in the ^1H NMR the integrals of the glycerol (starting material) are much bigger than the glycerol phosphates?
- Something similar happened with the nucleosides. Different from the glycerol phosphates, the nucleotides are well known signals in the ^1H NMR. For quantification in nucleoside phosphorylation is more realistic the use, for instance, for the purines the anomeric proton. For adenosine the chemical shift of the anomeric proton for the adenosine, 2'-AMP, 3'-AMP, 5'-AMP and the 2',3'-cAMP are well known (see: J. Am. Chem. Soc. 2023, 145, 23781–23793) so the quantification would be more accurate.
- One example of these discrepancies of quantification is figure S62. According to this figure, the authors claimed: “It seemed clear that MPG poorly reacted in the presence of cyanamide (65% unreacted MPG left after 5 days) when compared to urea (18% left after 5 days)” these % of conversions were measured using RP-HPLC and analyzed by ELSD, but table S27 shows that MPG phosphorylation using 1 eq of cyanamide or 1 eq of urea was 51.5% and 52.9%, respectively. The same plot shows that increasing the eq of urea the % increased but not mentioned at all what happens if the eq of cyanamide increases as well.
- Urea as an “activator”, promotor or just as a condensing agent with inorganic phosphate is very well known and has a history in prebiotic phosphorylation even has been intensely investigated. Many aspects for consideration:
- Based on the results presented reactions (e.g. Fig. S57, S60, S66, S82), cyanamide gave better conversion % than urea, so why don't pursue this agent that is less known? In the case of Struvite as a P source the difference was more than 20%, just for comparison.
- There is not any NMR (for supporting the claims) of the blank reactions (blank means just the alcohol or nucleoside and the P source, then the P source and the “Dehydrating /Condensing Agents/Liquidisers/Catalysts”). There are some experiments of urea (heating to see the degradation) but not with the P source.
- Some contradictory results. Example: Glycerol + Pi (no urea) conversion 35% and 3%, 115 °C and 75 °C, respectively. But then in the conclusion the claim was: “glycerol was phosphorylated even in the absence of any urea showing Pi conversions 36-77 % after 4-5 days at 115 °C”. Even worse, with cTMP without urea glycerol was phosphorylated about 90% (based on table S30), so why even bother to use urea or cyanamide if just glycerol and cTMP worked?

- MPG required the presence of urea to become efficiently phosphorylated at 115 °C, without urea at most 10% Pi was converted to organophosphates, but then (not with Pi but with cTMP the conversion of MPG without urea was 76% based on figure S68. Same comment, no need of urea if only MPG and cTMP worked.
- Section 5.1.2. It says: "The presence of equimolar amounts of urea (2a:2b), Pi and glycerol (5) in a neat mixture that was heated at 115 °C not only gave clear evidence for the accelerated degradation of urea to ammonium cyanate within the same timeframe" but based on the figure S42a after 24 h was clear the degradation, so why on the phosphorylation reactions the experiments were for 5 days?
- If the urea's degradation is clear after 24 h to ammonium cyanate, it would be a good idea to show the reaction e.g. glycerol + ammonium cyanate + Pi under the same conditions. This experiment would show the role of urea or the role of ammonium cyanate in the assisted-phosphorylation reaction.

Minor concerns:

- Figure 2 is not completely understandable. Orange exemplifies the Total conversion %, but only shows cyanamide and Pi and Struvite, while urea and cTMP, PPI and SPi results. Next, for the acyclic phosphates %, shows cyanamide and cTMP and Struvite, while urea and Pi, PPI and SPi. For the Cyclic 1,2-phosphates % cyanamide and Pi and Struvite, while urea with the rest. Finally, is it complex to see the full product distribution. The best idea will be divided into two portions the cyanamide and urea results, each one with the corresponding P source.
- Following the above, taking as an example glycerol: The data were taken from figure 2, so when the hyphen is there, it is because based on the figure that data is not shown, so it is complex to see the full total conversion. Also, the first box started with cTMP, next box (yellow one) started with Pi. There is no sequence for reading the data. It is not necessary to change completely the figure, just redraw in a way that will be easier to understand and to see the conversions.

Total Phosphorylation%	Acyclic phosphates	Cyclic 1,2-phosphates	Cyclic 1,3-phosphate
1 2 1 2 1 2 1 2			
Pi 97 -- 81 8 -- 4			
cTMP - 98 77 -- 3 - 11			
PPI - 85 - 74 - 6 - 5			
SPi - 28 - 28 - 1 - -			
Struvite 64 - 58 - 2 - 1 -			

- In the same figure, the part of the nucleosides, if the authors mixed A, G, U and C (the canonical nucleosides), what does the other (2%)? What another nucleoside can be?
- Why they used the terms "Heated neat blends"?
- Table S19 shows the reaction scale dependence of conversion of MPG, how they explained that using [13C]urea was a 100% of conversion? But when they used [15N2]urea was only 29% of conversion.
- "At the 0.5 mmol scale we detected that Pi consumption reached an end after 4 hours showing an exponential rate constant at 115 °C that was essentially the same as in the glycerol phosphorylation reaction", again why let the reaction for 5 days if after 4 hours all the Pi was consumed?
- Why in SI section 2.2 explained the procedure, showed the Fig S12, but then refer the Fig S54 – S56, why didn't show after this section?
- Figure S19a, can the authors explain why in the 31P NMR around 16 ppm there are at least 4 signals?
- Figure 1 shows Dehydrating / Condensing Agents/ Liquidisers / Catalysts, why the use of catalysts world?
- Only was used thymidine (for deoxyribonucleosides series), so that need to be in the figure 1.
- Figure S59, shows 1H-31P{1H} HMBC spectra, the X axis is not shown.
- It's a good idea to make some comments or a possible explanation of the differences in the product distribution between cyanamide and urea, example, why cyanamide sometimes gave more of the cyclic products than the acyclic ones?
- The figures S63 and S65 only show a window of the complete spectra, it would be good if the authors show the spectra from 20 to -25 ppm, since the signal of cTMP is around -22 ppm.
- The authors have proof of formation of the compound on figure S85, the 2',3'-cyclic adenosine monophosphate with the urea in the 5'- position?
- In the case of adenosine and guanosine phosphorylation, the easiest way to see the products is checking the anomeric region (around 5.8 and 6.4 ppm), figure S86 shows very poor conversion to the adenosine phosphates, while figure S89, shows good signals for guanosine phosphorylation. If the authors check this 1H NMR, they can identify the 2'-, 3'-, 2',3'- and even the 5'- phosphates products.
- For pyrimidines the easiest would be the aromatic proton of the ring core of uracil or cytosine.
- Figure S103 does not give any information, a good idea would be to compare the initial NMR of each individual nucleoside and then the spectra after the reaction. There are a lot of signals, and the graphic lost sense.
- It is confusing that figure S100 shows adenosine conversion 35%, but under the same conditions (urea and 1 eq 3a, 115 °C for 5 days), figure S122 shows almost 91% of conversion, how happened this?
- Figure S132 and the corresponding table S49 show the quantification of the reaction of the mixture of nucleosides + urea + Pi and 3a. There are a lot of signals, the quantification based on that figure is almost impossible. The data in my opinion is better just to show the NMR but not make any claim about the conversion.
- Something that is not clear why the authors chose valine? Why the relevance of this amino acid and not another one? Why only use valine for the nucleoside phosphorylation and not for the alcohols?
- The preliminary data of the vesicles are good, but I think that they can get deeper in these experiments and maybe try to write another manuscript. There is not a lot of information about this on the main paper, so it would be good to separate.

Overall, there are some interesting ideas presented herein; nevertheless, the thoroughness of the analytical/kinetic work (SI)

is not fully used in the main paper. Some experiments, ideas and conclusions that the authors wrote in the SI file would fit in the main paper. I recommend these matters be reviewed and as needed, experiments repeated under more stringent conditions (like the blank reactions, as well try to see if they can quantify using ^1H NMR) before the manuscript be considered for publication.

Version 1:

Reviewer comments:

Reviewer #1

(Remarks to the Author)

I have read the responses to my queries and I think the authors have made extended check for the computational section. The authors also answered at length the queries of the other reviewers. Therefore I am favourable to publish this work in this revised form.

Reviewer #2

(Remarks to the Author)

The authors have addressed and corrected the most important points. Thank you again for the detailed explanations. In my view, the manuscript has gained significantly in information and is acceptable for Nature Communications.

Reviewer #3

(Remarks to the Author)

Phosphorylation of alcohols and nucleosides is one of the most interesting and challenging topics in prebiotic context. In this paper "A scope of prebiotic neat reaction conditions and the mechanism of urea-assisted phosphorylation of glycerols, other alcohols and nucleosides", the authors promoted the phosphorylation of different compounds using urea/cyanamide and other "auxiliaries" to achieve the phosphorylation of glycerol, other alcohols and nucleosides.

Overall, the results are interesting and present a very detailed study with a lot (a lot) of information (some of that information is not even mentioned in the main paper), nevertheless, I would recommend publishing this manuscript, after reading the new version, the authors took into account most of the concerns, some other concerns they responded to the reviewers pdf, but overall, it is ready for publication.

AUTHOR'S RESPONSE TO :

REVIEWER COMMENTS

Reviewer #1 (Remarks to the Author):

This study definitively establishes how urea can assist the phosphorylation of alcohols and nucleosides. The work is comprehensive, employing a multitude of experimental techniques and a detailed SI that provides specific details. I was asked to provide feedback on the computational part as I am not an expert in experimental measurements. I have read the paper in its entirety and found the writing style challenging to follow. Many sentences attempt to convey too much information in a single location.

Thank you for mentioning this point. We tried to simplify long sentences by cutting them into several shorter ones.

In terms of the computational aspect, the authors analysed three PES, which are detailed in Fig. S13, S14 and S15 of the SI.

Actually, these are four PES two of which were shown superimposed in Fig. S13, and the next two PES were detailed separately in Figs. S14 and S15. We have changed this now.

Fig. S13 shows the pathways for associative phosphate dehydration with and without extra water molecules. Fig. S14 and S15 show a dissociative phosphate activation path without/with added extra water. All these PES have been characterised at the DFT level using the M06-2X functional, which includes London dispersion interactions. This is important as the considered molecular aggregates are relatively large and dispersion may act as a compacting factor. A modest 6-31G(d,p) basis set was adopted for all calculations (geometries, energies and thermodynamic corrections).

In the revised version, according to the suggestions made by Reviewer #1, we have recreated the 4 PES in three new Supplementary Figs. 13-15, where in Fig. 13 there is only one exemplary PES showing mainly the comparison of the methods M06-2X versus CCSD(T), and in Supplementary Figs. 14 and 15 are depicted 2 superimposed PES each, using one and the same method but comparing two basis sets, 6-31G(d,p) with 6-311++G(2d,2p), see response 1.1.

1. The "energetic" must be refined in terms of both the level of calculations and the basis set. All the considered PES envisage a negatively charged system, for which diffuse functions are essential. A single point energy evaluation for all stationary points and TS at M06-2X/6-311++(2d,2p)//M06-2X/6-31G(d,p) is the best approach. I recommend running a single point energy evaluation at CCSD(T)/6-311++G(2d,2p)//M06-2X/6-31G(d,p) for the reagent, first and highest TS, and the corresponding intermediate of the most promising path (Fig. S15). This will allow us to ascertain whether M06-2X is suitable for this type of rearrangement. A potential correction to the M06-2X for the whole PES can then be made.

1.1. We thank Reviewer #1 for this important comment. Geometry optimizations were performed at the M06-2X/6-31G(d,p) level of theory using Gaussian16 software. As suggested by the reviewer, a single-point energy calculation for the associative phosphate dehydration pathway (Pathway 1, without water) were performed at both M06-2X/6-311++G(2d,2p) and CCSD(T)/6-311++G(2d,2p) levels of theory. However, due to the extremely high computational cost of the CCSD method, the single point calculations were limited to this first pathway only. Unfortunately, our computational resources did not allow us to calculate all reaction pathways using the CCSD level method within a short time. It is well known that the cost of CCSD increases very rapidly with the number of basis functions. This limits its applicability to small systems, especially when many geometries must be computed. Bartlett & Musiał noted in *Rev. Mod. Phys.* **79**, 291 (2007), (<https://doi.org/10.1103/RevModPhys.79.291>) that despite its high accuracy, the CCSD method is rarely applied to large molecular systems due to its computational scaling and memory requirements. Therefore, the CCSD single point calculations were restricted to the P1-U3W0 pathway in order to balance accuracy and feasibility, and we used the DFT/M06-2X level systematically for all four PES. As shown in the energy profile in Figure here below (next page, also Suppl. Fig. 13), small basis sets such as 6-31G(d,p) may lead to a significant underestimation of transition state energies. This is due to

their restricted ability to describe electron correlation, polarization, and charge delocalization accurately, especially in transition states. This observation is reported in the literature, for example, by Frank Jensen in *WIREs Comput. Mol. Sci.* **3**, 273–295 (2013): "Smaller basis sets like 6-31G(d,p) often underestimate transition-state energies and barrier heights due to their limited flexibility." (<https://doi.org/10.1002/wcms.1123>). Moreover, Papajak et al. indicated in *J. Chem. Theory Comput.* **7**, 3027–3034 (2011) that the diffuse functions impact significantly the transition state energies, the long-range interactions, especially for systems involving partial charges or non-covalent interactions (<https://doi.org/10.1021/ct200106a>). Thus, the marked difference in transition state energies between the two basis sets can be attributed to the presence of additional polarization and diffuse functions in the 6-311++G(2d,2p) basis set, which allow a more accurate modelling of the electronic structure near the transition states. Note that the relative energies between intermediates remain stable, suggesting that the main effect of the extended basis is the underestimation of transition barriers without compromising the overall mechanistic profile. *These observations confirm that M06-2X/6-311++G(2d,2p) provides a more reliable energetic landscape, and the differences we observed are not artifacts but reflect a more accurate electronic description.*

Figure Supplementary 13. Associative phosphate activation pathway P1_U2W3 starting from one inorganic dihydrogenphosphate anion and three molecules of urea (U3) in absence of water molecules (W0) calculated at M06-2X/6-31G(d,p) (blue), M06-2X/6-311++G(2d,2p)//M06-2X/6-31G(d,p) (green) and CCSD(T)/6-311++G(2d,2p)//M06-2X/6-31G(d,p) (red) levels of theory. See Supplementary Fig. 14 for atomic details, relative Gibbs free energies in [kcal/mol].

We have added to Section 1.5 Quantum theoretical methods the following text and Suppl. refs. 8-9:

It is well established that density functional theory (DFT) predicts molecular structures and harmonic vibrational frequencies of substantially higher accuracy than obtained via Hartree-Fock (HF) calculations (Suppl. refs. 1-2). All calculations in this work were performed with the *Gaussian09* suit of programs (Suppl. ref. 3). In the present work, Truhlar's Minnesota functional M06-2X (Suppl. ref. 4) in combination with the 6-31G(d,p) basis set (Suppl. ref. 5) were used to fully optimise the geometries of reactants and products, and also were used to search for all possible geometries of transition states and intermediates. The M06-2X functional is highly nonlocal with double exchange (2X) and has been used successfully for a combination of main group thermochemistry, kinetics, and noncovalent interactions (Suppl. refs. 6-7). The performance of the method and basis set has been validated by the comparison to the energies of single points calculated with the most accurate CCSD(T) method (Suppl. ref. 8) and the more extended 611++G(2p,2d) basis set (Suppl. ref. 9): M06-2X/6-311++G(2d,2p)//M06-2X/6-31G(d,p) and CCSD(T)/6-311++G(2d,2p)//M06-2X/6-31G(d,p). The differences of single-point energies between

both methods using the extended basis set were very small; the CCSD energies were higher by 3.7-13.6 % for the intermediate states and 1.3-12.2 % for the transition states. Vibrational frequency calculations were carried out to ensure that the geometries obtained from M06-2X/6-31G(d,p) were indeed local minimal or saddle points on the potential energy surfaces and to determine the zero-point vibrational and thermal corrections to the Gibbs free energies. The Intrinsic Reaction Coordinate (IRC) (Suppl. ref. 10) calculations were performed to confirm the correct connections between reactants, transition states, intermediates, and products on the potential energy surfaces.

2. The captions for Figs. S13-15 miss the energies to be Gibbs free energy as also commented in the text.

1.2. Done. Thank you.

3. The addition of methanol (the simplest alcohol to be phosphorylated) at a specific point of the PES is an important and striking addition. This is not achievable in practice as the alcohol is always present and can interfere or play a role similar to extra water in the early stages of the PES. The authors' proposal simplifies the whole PES, and I understand their reasoning. However, this merits a comment and some hypotheses of the potential changes due to its presence from the very beginning.

1.3. This is a good remark and we thank the reviewer for this observation. Certainly, in the experimental prebiotic reaction the alcohol is present in the beginning of the reaction, similar to the residual water, which may be crystal water in the solid urea, whereas the experimentally tested alcohols were significantly larger than methanol, the smallest being glycerol. Therefore, there is the possibility for the alcohol molecule to indeed interact with the different intermediates and transition states in the early stages of the potential energy surface (PES). In our calculation, methanol was introduced at a later point of the PES. This choice was made in order to isolate the effect of adding methanol at a well-defined phosphate intermediate geometry. However, we fully agree that the introduction of methanol from the beginning of the reaction could affect the obtained energies by forming H-bonds. To this end and to that of response 1.1, we have now **added** in the revised Supplementary Information document *at the end of Section 3.1. State of the art and objectives concerning the reaction mechanism of the urea-assisted phosphorylation of alcohols* the following text (and **modified Suppl. Figs. 13-15**):

In all four pathways we assumed that the methanol molecule did only loosely interact with the reactants at the beginning of the potential energy surface. We checked this with a calculation of the optimised initial state in the presence of methanol and compared the energy with respect to the initial state in the absence of methanol plus the energy of a free methanol molecule. The changes were tiny, which supported our approximation to introduce the methanol at a later stage in the potential energy surface.

To ascertain a convincing strategy that would consist of a good balance between calculation time and accuracy of the results, we first focussed exclusively on the Pathway 1_U3W0 and compared methods and basis sets (Supplementary Fig. 13). Single-point calculations of the energies of the geometries that were found at the M06-2X/6-31G(d,p) level of theory were calculated using the extended 6-311++G(2d,2p) basis set, in order to allow to more accurately take into account electron correlation, polarization, and charge delocalisation of the anionic cluster. Next, we evaluated the performance of M06-2X against the CCSD(T) method using the diffuse 6-311++G(2d,2p) basis set. The obtained results showed that the energy differences between M06-2X and CCSD were relatively small. Hence, when combined with a sufficiently extensive basis, M06-2X provides an excellent agreement with high-level ab initio CCSD for the relative energies of stable intermediates. This validated our option to use the M06-2X functional to describe the aforementioned four pathways and compare the relative Gibbs free energies of the stationary states (intermediate and transition states) at the M06-2X/6-311++G(2d,2p) level of theory (Fig. 4 and Sections 3.2 to 3.4).

In **Section 3.3. Thermodynamic preferences and kinetic competition** we have also **added** a more cautious formulation in the second paragraph reminding of the above comments and responses 1.1 and 1.3:

[...] Second, the thermodynamics: like the diluted aqueous hydrolysis equilibrium state of phosphoric monoesters^{suppl. ref. 24}, the most stable 'final' product state in neat conditions is disfavoured in the organocatalytic phosphate dehydration, Pathway 1. Methyl phosphate, water and urea are

endergonic with respect to their corresponding initial state by about +11.6 (U3W0) and +6.7 kcal/mol (U3W2) on Pathway 1 (Suppl. Fig. 14). This is to be taken with a margin, since the effect of adding one molecule of methanol in the middle of Pathway 1 (tested to have a very small effect on the initial state) may not be completely nil. On Pathway 2, however, the molecular cluster containing methyl phosphoric acid and inorganic carbamate CA_i is favoured by -6.3 and -11.7 kcal/mol (to be taken with a margin, for the same reason). Hence, using our initial states (but see the note above) the dissociative phosphate activation pathway is not only kinetically favoured over the associative urea-catalysed phosphate dehydration pathway. The urea-assisted methylation of inorganic phosphate by methanol is also thermodynamically more favoured when urea degrades (Pathway 2) than when it is regenerated (Pathway 1).

4. If I have understood correctly, the authors mimic a situation in which water is almost absent.

1.4. Yes. More precisely, in the model that we call U3W0 there is no water present whatsoever, hence $P_i / \mathbf{2a} / H_2O = 1:3:0$, whereas in the model U3W2 the molecule ratio was $P_i / \mathbf{2a} / H_2O = 1:3:2$. Both models were calculated for both PES, termed "Pathway 1" and "Pathway 2", i.e., the "associative phosphate dehydration" and, respectively, the "dissociative phosphate activation" pathways. In the experiments, where glycerol (**5**) or MPG (**6**) were the alcohol, we measured the mol fraction of residual water (before starting to heat) and determined $P_i / \mathbf{2a} / H_2O$ (exp) = 1:1:0.2-3.6 (cf. Supplementary Table 20).

This means the reactions are carried out in an almost solid-like environment where the molecular mobility is reduced.

1.5. It was most often a mixed solid-liquid phase, because at this temperature (75°C or 115°C) at least the alcohol (except the nucleosides) was liquid. The solubility of urea in hot glycerol or hot MPG is not known. The solubility of sodium dihydrogenphosphate (P_i) in hot glycerol or hot MPG is not known either. When the alcohol was also solid (experiments with the nucleosides **11-15**), we added a small amount of what we called "liquidiser": liquid formamide (**3a**). In other words, almost always it was a mixed solid-liquid phase at least in the beginning of the reaction. Towards the end, that is, after 5 days of heating in an open vessel, the mixtures appeared to be more solid than liquid or entirely solid.

The DFT calculations instead leave full mobility during the geometry optimisation. Having forced some geometrical restraints will further increase the kinetic barrier. The authors must comment on this.

1.6. Introducing geometrical restraints will certainly increase the kinetic barrier, but this would require a modelling of the constraining environment. It is not sure that the environment will consist of (certainly not definite) solid arrangements of molecules whose composition may be made of water, phosphate by-products, urea, ammonium cations adding to Na^+ , etc. This is almost impossible to consider all possible arrangements that could be involved. One could introduce a continuum model like for a solvation, but that could not deliver a reliable energy shift.

To be more specific on point 4 raised by Reviewer #1, we added in the Supplementary Information in the second paragraph of **Section 3.4. Conclusion on the mechanism**:

We haven't systematically tested other urea- and water-containing initial states (UxWy). For example, we did not test initial conditions in which the number of P_i molecules would equal that of urea molecules (U1), albeit most experiments were carried out at an equimolar ratio. Neither did we model any environment made of molecules in an explicit solid state that would constrain the mobility, thus, further enhance the kinetic barriers of the reactive complexes, nor did we calculate the potential energy surface beginning with a complex where one of the urea molecules was already tautomerised to charge-neutral isourea right from the start. This could be the case when cyanamide was hydrolysed to first generate this rare isourea tautomer, before it would spontaneously tautomerise to urea ... (etc.)

To be clearer in the main paper text, with respect to both points 3 and 4 raised by Reviewer #1, we begin the **Conclusions for the geochemical context** as follows:

Our *ab initio* calculations have shown that urea's interactions with water and methanol are relatively weak when compared to those with inorganic dihydrogenphosphate (P_i). The dissociative phosphate

activation pathway (Fig. 4B) clarifies how P_i accelerates as an “acid-base catalyst” the degradation of urea to carbon dioxide and ammonia, viz. through catalysing the direct or water-relayed formation of a zwitterionic tautomer of urea predisposed to break one carbon-nitrogen bond and generate ammonium cyanate (NH_4NCO) (Supplementary Fig. 15). The potential energy surfaces give a rationale for how inorganic carbamoyl phosphate (CP_i) is generated in neat urea-phosphate ($2a + P_i$) mixtures through transitional ($NH_4^+ + NCO^- + P_i$) clusters (Int-3 in Fig. 4B and Supplementary Fig. 15), and how CP_i acts as a universal reaction intermediate with alcohols (Int-5).

In the more technical description of the theoretical results in the Supplementary Information we describe more precisely our findings about the above mentioned “acid-base catalysis” (= organic chemist’s jargon), by **modifying/adding** in **Section 3.2. Associative (organo-catalytic) versus dissociative (eliminative) phosphate activation :**

As can be seen in Supplementary Fig. 14, the associative pathway consists of the urea-provoked dehydration of phosphate, that is, a pathway where urea acts overall as a catalyst (is not consumed during the course of the reaction). The first transition state, in which one molecule of urea is deprotonated to the isourea anion $H_2N-C(O^-)=NH$ to immediately give carbamimidic phosphoric anhydride $HN=C(NH_2)OPO_2(OH)^-$ and one water molecule, is the key step that determines the reaction kinetics in, both, the U3W0 and U3W2 milieus, see Pathway 1 Supplementary Fig. 14 (unlike the ball-and-stick models, the key molecular structures are shown, for more clarity, without assisting urea and water molecules). This water molecule originates from a proton of the aforementioned catalytic urea molecule and a hydroxyl anion HO^- that was eliminated from the dihydrogen phosphate anion $P(OH)_2O_2^-$ transiently creating metaphosphoric acid $PO_2(OH)$, see [first transition state **TS-1**][#]. The recalculated Gibbs free energies of the stationary states shown in Supplementary Fig. 14 (and 15) are more accurate (dotted lines). The presence of two assisting water molecules (added right from the start) lowers the activation energy of the transition state **TS-1** by 35 kcal/mol (from 101 to 66 kcal/mol) and influences less the rest of the potential energy surface. The formation of metaphosphate and the regeneration of urea costs 22 and 36 kcal/mol activation. After adding a molecule of methanol, 21 (U3W0) and 25 kcal/mol activation barriers are needed for the methanolysis step in U3W0 and, respectively, U3W2. The final state containing all recovered urea molecules along with methyl phosphate and water has been found to be endergonic with respect to the initial state by $\Delta_r G^\circ \approx +7$ to $+12$ kcal/mol under U3W0 and U3W2 conditions (rightmost levels compared to leftmost initial Gibbs free energy level).

This associative dehydration pathway seems at a first sight too slow to be competitive with the dissociative phosphate activation pathway, as illustrated in Supplementary Fig. 15 denoted Pathway 2 (again the key molecular structures are shown without assisting urea and water molecules for greater clarity). In that second pathway, one of three urea molecules first tautomerises in an assisting-urea-mediated fashion to another, less costly zwitterionic isourea tautomer $HN=C(O^-)-NH_3^+$ which is more rapidly achieved than the phosphate-associated isourea anion on Pathway 1. In the U3W0 milieu roughly 46 kcal/mol less activation energy is spent when compared to that of the organo-catalytic Pathway 1. On Pathway 2 **TS-1_U3W0** needs 56 kcal/mol for a direct N-to-N migration of one proton on the same urea molecule. **TS-1_U3W2** needs a mere 32 kcal/mol for this tautomerisation which involves a water molecule that relays the proton from one N atom to the other N of the same urea molecule (compare the leftmost orange with leftmost blue shadowed boxes in Supplementary Fig. 14). This zwitterionic isourea tautomer is predisposed to dissociate into ammonia NH_3 and isocyanic acid $HNCO$, which costs another 14-19 kcal/mol activation in the absence or, respectively presence of assisting water molecules and sums up to an overall trajectory-determining transition state of roughly 37 kcal/mol in the U3W2 milieu (**TS-2** in Supplementary Fig. 14) compared to 60 kcal/mol on the corresponding organo-catalytic pathway.

I will be happy to review again this section of the work in the revised version.

1.7. We thank Reviewer #1 and hope to have satisfied all points raised above.

Reviewer #2 (Remarks to the Author):

The manuscript entitled 'A scope of prebiotic neat reaction conditions and the mechanism of urea-assisted phosphorylation of glycerols, other alcohols and nucleosides' summarizes the results from phosphorylation experiments with model alcohols under various reaction conditions.

Here are my comments:

In the first sentence of the manuscript the authors make already a quite bold statement: 'Two theories of the origin of life on Earth, one located in the Hadean seafloor, the other on the surface of subaerial landmasses and basins, need reconciliation.' I do not agree with this.

2.1. In the abstract we do (can) not cite the literature references on which we base our statements. Also, there is a restriction (for us in this paper quite strongly limiting) of a maximum of 70 literature references in the main paper. For this reason, we needed to resort quite a few citations into the Supplementary Information document. We decided therefore to 'shift' the introduction to quantum chemical methods for phosphorylation reactions, as well as the discussion of nucleoside phosphorylation, which both include a lot more citations than in any other subject of this work, into the SI. But we kept in the main paper all necessary citations that concern the geochemistry results on which we base our statement in the abstract, in the introductory "geochemical framework" (with citations), and in the last paragraph of the main text (with citations). We do not find the theories "bold" but objectively based on scientific published papers, for instance refs. 4 (Walton et al. 2023), 5 (Russel & Ponce, 2020), 7 (Toner & Catling 2020), 8 (Walton et al. 2021), etc., but in particular, what regards the two above mentioned theories, refs. (numbering in the revised manuscript) 20 (Camprubí et al. 2019), 22 (Damer & Deamer, 2022), 28 (Brady et al. 2022), 67 (Ross & Deamer 2019), 71 (Westall et al. 2018), 72 (Martin & Russel, 2017), 73 (Konn et al. 2015), 74 (Nitschke et al. 2024), 75 (Dobson et al. 2000), 76 (Tuck, 2002), 77 (Jordan et al. 2024) and 78 (Geisberger et al. 2023).

In the last paragraph of the main paper, where we elaborate on what has been stated in the abstract, and there we cite refs. 20, 28, 67-78: "Our results should provide valuable data for the theory on the outbreak of life in seafloor hydrothermal-sedimentary systems⁷¹ or deeper hydrothermal-vent systems^{72,73} that offer dielectric barriers as compartments for efficient chemiosmotic work to be casted⁷⁴ albeit, as such, with a limited evolvability potential." Etc. (see below).

Firstly, I would not call different (reaction) conditions a theory

2.2. These are two "schools of thought", if you wish, or "camps". One camp is convinced that life started in the Hadean seafloor, the other camp puts the beginning on the surface of subaerial landmasses and basins. Both camps have nicknames, for example, "Metabolism first" and "Replication first", or "Bioenergetics" and "Molecules", and the like. These theories have their protagonists and, like in other scientific areas, the protagonists do not communicate much with one another. Both camps are represented by researchers who have their own techniques of measuring and, naturally, their own way of concluding. The group of authors of this manuscript are chemists who traditionally work in the lab and, thus, are as if innately (automatically) associated with the "Replication first" camp: we synthesize organic molecules, test their properties and reactivities, and conclude accordingly. Two of us, though, are geologists (Isabelle Daniel), at least in part (Anastasiia Shvetsova, the first author), who do have ties with colleagues of the other camp. We should like to propose a "truce" to both camps, or at least pave the way for it.

and secondly, why do these spaces have to be brought together?

2.3. Because we argue in the following way, and this is our opinion that we would like to publish, thus, bring into discussion of the interested scientific community (last paragraph): "We should think of the possibility that probably neither of these initial subaerial and seafloor systems could build up sufficient complexity to become alive on their own. The subaerial locations lacked congregated long-term energy sources to unflinchingly push forward the disequilibrium to a further level of complexity, and the seafloor was not a locus apt to amass sufficient organic phosphates from the elimination of water out of their precursor molecules."

I think it is only due to our limited imagination that we are not able to accept different conditions. It is interesting to see that many find it difficult to accept different reaction conditions. Even on our planet there are such large temperature differences, pressure differences, if we look at the oceans or volcanism. This needs to be rephrased.

2.4. In our work, we simply distinguish two different classes of conditions: *i)* arid, no solvent or not much, sometimes we call them 'dry', or else 'wet and evaporating' or 'wet-to-dryness', which we all link to the scarce subaerial landmasses of the Hadean era. *ii)* The other is highly diluted or dilutable, under pressure, always aqueous and heated and/or provided with chemical energy 'from below', which we link to 'hydrothermal conditions' as is described in many of the above-mentioned papers (refs. 20, 28, 67-78), also others cited in other places of the main text: refs. 25-27 (Schreiber et al. 2012, Mayer et al. 2015, Shikuya & Takai 2022).

We test in a very comprehensive way the first arid class of conditions, with respect to the phosphorylation of alcohols, and we put our results at the end of the main text in relation to what is known from many papers on hydrothermal conditions. In those, no comparably efficient phosphorylation reactions are known, which is readily explained by the unfavourable thermodynamics in such fully aqueous conditions (under whatever pressure).

Ultimately, the authors themselves recognize the dilemma: 'However, the lately growing number of different prebiotic dehydrating conditions proposed to have furnished crucial organic phosphate esters shortly before the first living cells have emerged in the late Hadean and early Archean eras, leave at present the scientific community with a very large range of possibilities that lack thus far appropriate experimental comparison.' Therein lies the mistake, you don't have to compare the conditions but investigate whether such reactions are possible under given conditions. There are very good studies in literature that do so, even if some researchers immediately criticize it if there is no 'high yield' reaction.

2.5. We do not "criticize" low yields, we publish ourselves low conversions of inorganic phosphate sources into organic esters (of monools). But we respectfully disagree with Reviewer #2 in being, or not, faced with the necessity of comparing conditions. This is not a mistake, we do it on purpose. We think that, indeed, the concerned scientific community lacks truly comparable experimental data, this is one major reason for this comprehensive work. Many colleagues just carry on for decades to find conditions that give 'good' or 'not so good' results, mainly in the context of the theory, or camp, to which they feel adhered. But far too few actually truly compare their data with that of others, for the simple reason that such differently conducted experiments are very difficult to compare in a convincing way, and even more so, convincing for a protagonist of the other camp.

What the geochemist needs to know from the organic chemist, we feel, is precisely how all these various reaction conditions, that are being continuously discovered and proposed usually by chemists, actually do compare with one another. To be able to truly compare, one first needs to select a given set of plausibly prebiotic reaction conditions and apply them on a set of representative starting molecules, this is what we did here. Next, one needs to not only determine and compare apparent rate constants (which we did) but also start using highly complex mixtures that actually might have taken place on the early Earth. We started in this work with quite complex mixtures that contained an amino acid or a mixture of alkanolic/fatty acids, glycerol and ribonucleosides and report interesting, unexpected and unseen thus far results.

Of course, one should continue to test even more complex mixtures. We had to stop where we stopped for obvious reasons, analytical, time, funding.

The following papers should be cited in the context of oxygenated precursors:

Beyazay, T. et al. Ambient temperature CO₂ fixation to pyruvate and subsequently to citramalate over iron and nickel nanoparticles. *Nature Communications* 14, 570 (2023).

Peters, S., Semenov, D. A., Hochleitner, R. & Trapp, O. Synthesis of prebiotic organics from CO₂ by catalysis with meteoritic and volcanic particles. *Sci. Rep.* 13, 6843 (2023).

2.6. In the revised version of the main paper, and if the Editor will allow us to do so, we could indeed cite these very interesting papers (now refs. 57 and 69) and enlarge the reference list to a total of 78.

Interestingly, the authors consider dissolving insoluble phosphates with amines or nitrogen compounds. Chemically speaking, this makes no sense.

2.7. We did not consider to “dissolve” inorganic phosphates with any amines, only chemically activate, or just physically mobilise, inorganic phosphates with urea, cyanamide, formamides and acetamides. One of our initial arguments was that, in the late Hadean era, not much amines could accumulate and be locally available yet. Simple ammonium cations are volatile over long (geological) time periods, such cations eventually evaporate as ammonia or methylamine and leave the rest of non-volatiles without amines. All amines, we argue, that were available and could appear accumulated under these conditions, in the hot subaerial “scenario” that we evoked, were amino acids (represented by, L-alanine, L-valine and D-valine), and nucleosides, thus, aromatic N-heterocycles being, chemically, tertiary amines, amidines (C6 of adenine, C4 of cytosine), ureas and lactams (C2 and C6 of guanine, C6 of hypoxanthine, C8 of xanthine, C4 of uracil and thymine) represented as purine and pyrimidine nucleobases bound to ribosyl or deoxyribosyl residues.

However, these “reduced-nitrogen molecules” (amino acids and nucleosides) were not sufficiently abundant, and do not bear a sufficient chemical potential, to play a role as “activating agents”, nor are they liquid and physically mobilizing — they were prebiotic building blocks, not less and not more. The less or non-volatile cyanamide, urea, formamide, acetamide and their N-methyl derivatives (**1-4c**) could have possibly been available in large excess amounts that would allow for a continuous, uninterrupted chemical (stoichiometric) activation of inorganic phosphates that would then react with the less abundant and more stable building blocks. Only large excesses of such high(er)-energy molecules could fight against the thermodynamics of hydrolysis, as soon as water was present for longer time periods.

Why not acids? Moran has shown that acids are also formed, even complexing acids. They even dissolve apatite. This is known from lemonades containing citric acid, which damage the teeth.

2.8. We did test several different acids, three molecular kinds: Pyruvic acid (**16**) and the fully saturated C10, C11, C12 and C13 alkanic acids **7abcd**, also sodium glycerylphosphate (**5b**) and sodium 1-dodecylphosphate (**9b**). None of those gave phosphorylation results of the eyebrow-raising kind. On the other hand, our experiments led to the discovery, to our knowledge for the first time, that pyruvic acid, rather than become phosphorylated, could have been a prebiotic precursor of a new class of polyketides (see Supplementary Fig. 196).

With regard to the phosphorylation of amino acids and other compounds, it would be advisable to search the chemical literature and not just the geochemical literature (in part in French(!)):

Rabinowitz, J. Recherches sur la formation et la transformation des esters lxxxiii [1]. réactions de condensation et/ou de phosphorylation, en solution aqueuse, de divers composés organiques à fonctions OH, COOH, NH₂, ou . Helv. Chim. Acta 52, 2663-2671 (1969).

Rabinowitz, J. Note on the role of cyanides and polyphosphates in the formation of peptides in aqueous solutions of amino acids, at room temperature, as a possible prebiotic process. Helv. Chim. Acta 54, 1483-1485 (1971).

Hatanaka, H.& Egami, F. The formation of amino acids and related oligomers from formaldehyde and hydroxylamine in modified sea mediums related to prebiotic conditions. BCSJ 50, 1147-1156 (1977).

2.9. Merci beaucoup! Again, in the revised version of the main paper, and if the Editor will allow us to do so, we could indeed cite these pioneering papers (now refs. 63-65, in addition to the latest ref. 66 by Šponer et al. 2024), thus, enlarge the reference list to 78.

If, however, we are not allowed to enlarge our reference list that much, we shall insist on all references of our previous list, which needs to be enlarged with 2 refs. that were previously placed in the SI (suggestion of Reviewer #3, see below). In that case, and depending on the Editor, we cannot keep refs. 63-65 and perhaps even not the refs. of your first two suggestions, now refs. 57 and 69 (Beyazay et al. 2023, and Peters et al. 2023, see above). This would be highly regrettable, though.

I also noticed that phosphorylation with small organic molecules is not mentioned at all. Considerable selectivities have been reported here recently. This applies in particular to imidazole derivatives, imidazolidine-4-thiones and 2-aminooxazole.

2.10. In the main paper, we have cited ref. 13 (Ter-Ovanesian et al. 2021) and ref. 14 (Yi et al. 2022). Both describe the carbamoylation of the amino group of α -amino acids with inorganic carbamoyl phosphate, and cyanate. We have cited ref. 15 by Maguire, Smokers & Huck, 2021, which is an important study carried out with in-situ-generated inorganic carbamoyl phosphate catalysed by imidazole to phosphorylate (in water) hydroxy acids, hydroxyl-bearing amino acids, glycerol and related 1,2-diols, as well as adenylates. We have also cited ref. 44, Fernández-García, Coggins and Powner 2017, where the role of, for instance, 2-aminooxazole together with inorganic phosphate are reviewed in the synthesis of pyrimidine nucleosides. In the main paper we are restricted in the number of cited references, so we have made our choice. Regarding the use of imidazolidine-4-thiones for the regioselective phosphorylation of nucleosides, indeed, we have now added ref. 43 (Bechtel et al. 2022) to the supplementary discussion of the nucleoside phosphorylations in Section 7.5.8 of the revised Supplementary Information, and mentioned their high 5'-selectivity. Thank you for reminding us of this *oubli*. We have found yet another new paper in this topic (Suppl. ref. 45), and more on nucleoside phosphorylations using cTMP. Therefore, **new** in Section 7.5.8. of the revised Supplementary Information:

The relative proportions and amounts of phosphorylated ribonucleosides obtained at 115 °C are comparable to those obtained by others from reaction mixtures heated up to 85 °C for several days in a urea-ammonium formate-water eutectic, liquid sulphur dioxide or subjected to 'dry-wet' thermal cycling using large excesses of both urea and phosphate (10-16 mol equivalents), also more reactive phosphate sources richer in protons from ammonium ions (Suppl. refs. 21, 39), and those in the presence of minerals, catalytic metal-doped clays, photoredox-active cerium phosphate, organo-catalytic 2,5-dimethyl imidazolidine-4-thione or in an initially acidic supercritical carbon dioxide-water (scCO₂-H₂O) two-phase environment (suppl. refs. 40-45). In contrast to the nucleoside mixtures containing catalytic imidazolidine-4-thiones as in-situ phosphorylating agent (suppl. ref. 43 [Bechtel et al. 2022]), or those with 20 mol equivalents urea and 3 mol equivalents NaH₂PO₄ or hydroxyapatite heated at 94°C in scCO₂-H₂O, where 5'-nucleotides dominated followed by carbamoyl nucleosides and acyclic 3'- and 2'-monophosphates (suppl. ref. 45 [Tagawa et al. 2024]), our hot neat solid-liquid conditions always produced a high fraction of 2',3'-cyclic phosphates (Supplementary Fig. 131), and expectedly higher than that found in the presence of borate (Suppl. ref. 46). Interestingly, [etc. ...]. Of note, proton-decoupled ³¹P{¹H} NMR spectroscopic evidence for the condensed nucleoside di- and triphosphates (Suppl. ref. 48) could not be found in the crude mixture containing all four nucleotides (no δ_P -doublets and - triplets for NDP = ppN or NTP = pppN). Lowering the reaction temperature to 75°C, replacing urea (**2a**) with cyanamide (**1**), or P_i with cTMP or SP_i brought nucleoside phosphorylation virtually to a halt, although pyrimidine nucleotides formed from **1** and P_i to some extent. As shown by others, nucleoside phosphorylation with cTMP is comparably weak, and requires dry-wet or hot-cold cycling in the presence of water where sufficient magnesium or transition metal ions are mandatory (Suppl. refs. 49 [Ozawa et al. 2002], 50 [Cheng et al. 2004]).

With regard to the selection of alcohols, even more interesting representatives would come to mind later on, as lipids come rather late in the chemical chain.

2.11. We are not so sure whether lipids came actually late in the "chemical chain", because more and more studies are being published on the prebiotic synthesis of alkanolic acids (see for example ref. 78, Geisberger et al. 2023 and refs. therein). We have tested the proposed alcohols, prebiotic precursors of phospholipids and nucleotides including the monools **7-10** and pyruvic acid (**16**), and have based our choice of the monools on arguments that have been now shifted from the Supplementary Information (suggested by Reviewer #3, see response 3.45).

Please correct the reaction scheme leading to benzoic acid in Fig. 4. A. It looks strange when benzoic acid is formed from other molecules out of the blue. The same holds in part for Fig. 4.B.

2.12. Benzamide and benzoic acid do not come "out of the blue" in Figure 4 (A and B). We do show the formulas of the reagent of each trap. For Trap 1, the "Nucleophile trap" (PhCOCl = benzoyl chloride), for Trap 2, the "Electrophile trap" 1. PhMgBr (Phenyl Grignard), 2. HCOOH (followed by the acidification of the sample).

I would rename the title of Figure 4 and also in the text. It is only a mechanism in part (TS in Fig. 4. A), it is more a proposed reaction cascade.

2.13. Figure 4 A and 4 B show the results, in a much-abbreviated form, of the DFT-calculated potential energy surfaces (PES) of both theoretical mechanisms, thus, the most important core structures of the calculated clusters (without assisting urea and water molecules) labelled with the relative Gibbs free energies of the corresponding stationary states (now recalculated and redrawn in Suppl. Figs. 13-15, see responses 1.1-1.3 to Reviewer #1), not just a cascade. Figure 4 C does show a cascade of molecules produced in the order that we have experimentally determined (as detailed in Section 5 of the Supplementary Information). We name these “identified experimental products”.

If the authors really investigated the mechanism of the reaction experimentally, I would expect the reaction orders for the reagents as a minimum requirement. Does it agree with the proposed model derived from the DFT calculations?

2.14. It is difficult to experimentally determine the reaction orders of the reagents of a reaction that is carried out in neat conditions, that is, without solvent. Therefore, no concentration dependence of the apparent rate constant could be interpreted. Indeed, we have performed only a limited number of different stoichiometries in our experiments. For the kinetic experiments, where we have determined the apparent rate constants of urea degradation and the consumption of inorganic dihydrogenphosphate (Fig. 3), we used only an equimolar stoichiometry. To reliably determine the experimental order of the reagents, we ought to carry out similar measurements at several different relative stoichiometries, best at different reaction temperatures. Then, however, we would risk to bias the actual reaction mechanism, being operative at a quasi-equimolar stoichiometry, with perhaps other mechanisms becoming significant (competing) in mixtures where one of the reagents is present in a strong excess or much depleted. To determine a reaction order from our equimolar dataset, the method of trying to fit functions having the reaction order (exponent to k) as a free fitting parameter other than 1, rather than the exponential functions shown in Supplementary Table 9,* would not be nearly precise enough to be able to claim an experimentally determined reaction order of the reagents. Given the mixed solid-liquid-phase nature of our neat mixtures, we were faced with quite large batch-to-batch deviations especially at different scales of the reaction, but not only (see Supplementary Figs. 11 and 19). In other words, the lack of homogeneity in such mixed solid-liquid phases even limits the precision of the determined apparent rate constants, see error margins in Fig. 3 for the exponential fittings, let alone that of reaction orders.

On the other hand, we based the initial conditions of our DFT calculations on what was already known from older mechanistic studies. A simplified version of the “associative phosphate dehydration” mechanism, as shown Fig. 4A and Supplementary Fig. 13-14, has been “very tentatively” suggested by Orgel & Lohrmann in 1974 (ref. 43) and later propagated (unproven) by other authors to be operative in water, whereby a hypothetical rare tautomeric form of dihydrogenphosphate was evoked (Supplementary refs. 20-22 : Burcar et al. 2016, Fernández-García et al. 2017, and Xu et al. 2017). In our calculations we have tested the existence of such a rare tautomer (two protons on one and the same oxygen atom of dihydrogenphosphate) and failed to identify with our method and basis set any transition state (single-imaginary vibrational frequency) that would lead from the normal tautomer of dihydrogen phosphate (OH twice) to the aforementioned hypothetical rare tautomer (O^- , OH_2^+), even though dihydrogen phosphate was surrounded by three potentially stabilising urea molecules and irrespective of the presence or absence of two additional assisting water molecules. This tautomer simply cannot exist (see second paragraph of Section 3.1 in the Supplementary Information). Luckily, we did indeed find a potential energy surface that led to the “associative phosphate dehydration” of the normal tautomer of dihydrogenphosphate (**Int-1** in Fig. 4A and Supplementary Fig. 14), albeit at a very high Gibbs free energy of activation (**TS-1** in Fig. 4A and Supplementary Fig. 14).

The “dissociative phosphate activation mechanism”, as shown in Fig. 4B and Supplementary Fig. 15, has been thoroughly elucidated for aqueous solutions by Shaw & Bordeaux 1958, by Allen & Jones 1964, and well completed — auf Deutsch! — on the relation between urea and inorganic carbamoyl phosphate by Seel & Schinnerling 1978 (Supplementary refs. 17-19). The second part of both mechanisms, which is the alcoholysis of a reactive intermediate generated from dihydrogenphosphate, or in the other sense, the hydrolysis of diverse phosphate monoesters, has been extensively described by Westheimer 1981,

and by quantum-theoretical methods in the group of Warshel and co-workers Kamerlin and Åqvist (Supplementary refs. 13-16). In that sense, our proposed models do agree with all these studies.

* To be more transparent with respect to the error margins for the “derived parameters” (apparent rate constants k and half-lives $t_{1/2}$) given in Fig. 4 of the main paper and in Section 5 of the SI, we have added a (new) footnote to Suppl. Table 9:

...

Derived parameters*	$y_{max} = A$	$y_{max} = A_1 + A_2$	$y_{max} = A_g$
	$t_{1/2} = \frac{1}{k}$	$t_{1/2n} = t_n * \ln(2)$ $k_n = \frac{1}{t_n}$	$k_d = \frac{1}{t_d}$

* The \pm values reported by Origin™ for derived parameters k and $t_{1/2}$ (Fig. 4, Suppl. Tables 10 and 18, Section 5.4) represent their standard deviations calculated from the covariance matrix during nonlinear least squares fitting, based on how sensitive the model is to changes in the data and indicate how precise the estimates point to the original data points.

See more details at https://www.originlab.com/doc/Origin-Help/NLFit-Theory#How_Origin_Fits_the_Curve

DFT calculations are mentioned, also the cartesian coordinates in the SI. Please provide information about the software used and the basis set, functionals etc.

2.15. We did provide this in Supplementary Section 1.5 (Quantum theoretical methods) and cited Supplementary refs. 1-10. Because we have now extended our calculations on a more elaborate diffuse basis set, as asked by Reviewer #1, we have redrawn Suppl. Figs 13-15 and added some new text in Section 1.5 (see response 1.1).

The diastereomeric interactions are indeed surprising.

Please check the references. I have noticed that certain references are missing in the text.

2.16. Done, thank you.

For me, there are too many open questions to make such conclusions from the experiments (Our results should provide valuable data for the theory on the outbreak of life in seafloor hydrothermal-sedimentary systems). I think that's completely exaggerated.

2.17. It is our opinion based on many publications and the corresponding author's decades-long experience of experimental work in the area, in addition to the ones that we are reporting here. As stated in our first response 2.1 to Reviewer #2, our work is a comprehensive in-depth assessment of the phosphorylation of lipid precursors and nucleosides under conditions that dominated on Earth's earliest solid, hot and mostly dry surfaces. However, we do not wish to conclude from our results that 'our camp' is right and the others are wrong. Quite on the contrary. In the manuscript, we have strongly condensed our results and conclusions in such a way, that we think should be attractive to read as a member of the metabolic camp, who are convinced that everything that happened on the early Earth, which led to the first living organisms, was under the first global ocean, in water of course. In this paper we are confronting the interested and concerned scientific community with our statement to join forces, to become more involved in collaborations and mutual points of view between camps, in order to overcome this barrier that is, in our opinion, more of a hindrance than a door-opener to reach an even deeper knowledge on the origin of life.

Because of these points, complete revision is necessary, and I do not consider the results to be as new as they are presented here.

2.18. We did completely revise the manuscript and Supplementary Information, and hope that our explanations to Reviewer #2 have helped to provide for a better opinion of our work than initially. We thank Reviewer #2 for his or her engaged input and the suggestions to expand our list of literature references (cf. responses 2.6 and 2.9).

Reviewer #3 (Remarks to the Author):

Phosphorylation of alcohols and nucleosides is one of the most interesting and challenging topics in prebiotic context. In this paper “A scope of prebiotic neat reaction conditions and the mechanism of urea-assisted phosphorylation of glycerols, other alcohols and nucleosides”, the authors promoted the phosphorylation of different compounds using urea/cyanamide and other “auxiliaries” to achieve the phosphorylation of glycerol, other alcohols and nucleosides.

Overall, the results are interesting and present a very detailed study with a lot (a lot) of information (some of that information is not even mentioned in the main paper), nevertheless, I would recommend publishing this manuscript, but the presentation of the results can be improved if they clarify the reasoning of some experiments, some statements on the discussion and finally the final conclusion.

Major concerns:

- Reading the main paper is a difficult task, there is not any specific order, for instance in the results section immediately, they refer to figures S142, Fig. S195 (but the SI finished in Fig. S194). Then mentioned Figs. S157 and S160-S163, Tables S55 and S56. In my opinion the manuscript needs to be written in a sense that it would be “easy” for the reader, not only for the understanding but for the flow of information.

3.1 It is true that we decided to write the very strongly compacted form of description of our work in the main paper in a somewhat different order, as if in a summarising state of mind serving a slightly different purpose than the strictly systematic order of the description in the Supplementary Information document. In the revised version, following the encouragement of Reviewer #3 concerning his or her but last comment and our response 3.45 (see below), we shifted more of the discussion shown in the Supplementary Information document into the main paper, and we tried to stay in a more chronological order of the main paper that should be better synchronised with that of the Supplementary Information.

- Normally, for a SI file the authors should ensure that the information is completely clear and well presented, but it is difficult to follow.

3.2 It is important to see that we do not necessarily follow the logics (the expectation) of an organic chemist who wants to have the starting molecules in the main focus. Our main focus is to group the whole into the most geochemically plausible reaction conditions, and only after that into the starting alcohols that are best phosphorylated, then second best, and in the end the weakest (in being phosphorylated). Therefore, we begin with urea-assisted reaction conditions despite urea having the thermodynamically weakest chemical potential as an activating agent. It is precisely the relatively weak thermodynamic potential of urea that makes it the most likely molecule to be abundantly available for quite a long period of time. All the other very potent agents, such as inorganic thiophosphate, trimetaphosphate and cyanamide will hydrolyse easier than urea, given an even very short geological time scale. Thiophosphate and trimetaphosphate or, for that matter, any other inorganic condensed phosphate will hydrolyse much more readily than urea to give inorganic orthophosphate (P_i). Cyanamide will slowly but steadily produce urea. So, we begin our story (the flow of information) with the “urea- P_i couple”.

To make this important point clearer in the main paper, we have added a few sentences to the **Conclusion for the geochemical context:**

[...] Abiotic urea consists of a thermally stabilised solid form of reactive ammonium cyanate that, by irreversibly evaporating ammonia and leaving cyanate to react, actually drives the phosphorylation reaction at temperatures close to that of boiling water and above (and below). Even at a mere equimolar ratio of alcohol and phosphate, urea can act as an effectual chemical ‘fuel’ of phosphorylation that will ‘burn’ anyway, that is, hydrolyse in the presence of water to produce ammonia and carbon dioxide. Therefore, out of the here tested dehydration and phosphorylation agents, urea in combination with P_i are geochemically the likeliest environment for prebiotic phosphorylations of simple organic precursor compounds, precisely because they bear the lowest chemical potential of all: solid urea- P_i mixtures could have acted on organic molecules for the longest periods of time before hydrolysing. [...] Chemically internally activated inorganic phosphate sources, such as condensed phosphates P_{ni} ($n = 2, 3, \dots$) generated by heat (trimetaphosphate, **cTMP**) or crystallised in minerals

(canaphite), and thiophosphate (SP_i) present in hydrogen sulfide-containing acidic volcanic waters during hours, days or longer¹⁹, can indeed phosphorylate alcohols, but not as long-lasting as urea and inorganic orthophosphate ($2a + P_i$). It seems that, over longer episodes including aqueous wet periods, the condensed P_{ni} were mainly hydrolytic reservoirs of P_i . In an alkaline milieu, however, cTMP is a potent and valid phosphate source, particularly, in contrast to the ($2a + P_i$) couple or SP_i that actually profit from available protons and slightly acidic conditions. Only cyanamide (**1**) has a higher chemical potential than cTMP (Fig. 1). cTMP at 115 °C is able to form cyclic carbonates from atmospheric CO_2 and 1,2-diols present in diglycerol phosphates (Supplementary Figs. 66-69). Carbonylation stabilises such organic phosphate esters and, as mentioned before in the context of hot urea + P_i blends, makes them more resistant against degradation over longer periods of time. However, ribonucleosides and monools are hardly or not at all phosphorylated by cTMP in alkaline neat conditions without abundant magnesium or transition metal ions. Both, SP_i and cTMP degrade to P_i when the pH drops, which is a key argument in the context of an atmosphere that bore huge amounts of carbon dioxide that would importunately acidify the waters by hydrolysing to aqueous carbonic acid⁶⁰. Therefore, the best immediate prebiotic value of cTMP is its role as a persistent condensing agent based on its efficiency in creating peptides from amino acids^{44,61,62}, and that of SP_i could be drawn in particularly alkaline environments, where the urea + tribasic P_i blend is an unreactive phosphorylation agent while tribasic SP_i still is (section 7.4.1 and Supplementary Figs. 74-78), and from its photochemical reducing power¹⁹.

The sections are mixed, like first experiments with glycerol and MPG and urea, then change to glycerol alone then move with cTMP with both and finally with the others P sources.

3.3 In the Supplementary Information document we begin with Section 1) Materials and Methods, where we now have much better explained the way, how we quantify our experimental results (an important point raised by this Reviewer, see later). We continue with 2) the design of experiments (the experimental setups), 3) the quantum theoretical work which sets the grounds of the priorities of the experimental sections that follow. We have renamed Section 4 into “Identification of the reaction products ... (new, without “and quantification”) to be clear that this is not yet the results section. In Section 4 we indeed do show glycerol and MPG only, but this is because we wished to explain, how exactly we determined the molecular structure of those products that delivered the most complicated and abundant crude reaction mixtures, expectedly from neat urea- P_i reaction conditions. We carry on with Section 5) where the kinetics of this reaction is studied, followed by Section 6) where we include gaseous volatiles and ¹⁸O isotopes, both in order to give better experimental grounds for the mechanistic study based on quantum theory alone (Section 3). So, in the Sections 4, 5 and 6 we use glycerol and MPG (our “star alcohols”) and the neat urea- P_i couple only as representative examples, not yet systematically. We could have chosen the nucleosides, or some of the monools, to explain the way how we identified our products, but we preferred to take glycerol and MPG.

Only then, in Section 7 we begin with the results that we name (as in the title of the main paper) “a scope”. There we start again with glycerol, continue with MPG, then the nucleosides, the monools and pyruvic acid.

The main paper, serves a slightly different purpose in the sense that we wish to attract first of all the geochemist reader, who does not want to first read all the systematic procedures such as sample preparation, quantification, identification of our products. Therefore, we follow a somewhat different order. After presenting the molecules in Fig. 1, we offer a “nutshell” paragraph where the most important experimental results are presented in an extremely compact form, although we do try to keep the order as in the SI, but not so strictly anymore (in this nutshell) and illustrate all experimental results with this “maximum/representative conversion” graphs in Fig. 2. We then follow with a briefing on the kinetic results (Fig. 3), naturally on glycerol and MPG only. Only then (having briefed what worked best) we pose the question about the mechanism of urea-assisted phosphorylations, which has been explored as described in the Sections 3, 4 and 5 of the Supplementary Information, and illustrate this in an extremely compact form in Fig. 4. We chose this order of the first part of the main paper (up to Fig. 4) on purpose, having in mind another kind of reader of the main paper and a different expectation, in other words, a much broader readership that needs to “get to the point” (their expected point) fast. What

follows in the main paper is the discussion section. We have sub-divided the discussion in the same order as in the Supplementary Information beginning with Section 7.

It would be great if they can write the sections based on the alcohol or nucleoside. It means, all experiments with glycerol, then with MPG and the rest of the alcohols (7, 8, 9 and 10), and finally with the nucleosides.

3.4 Again, we wished to order the discussion of the main paper, as well as the order of Section 7 in the Supplementary Information, following the let's call it the "kinetic competition" in a natural geochemical and prebiotic environment, where all these molecules could have accumulated in more or less the same location, in an arid environment on an island of the late Hadean era. We describe the reactivity of our starting alcohols that way (best to weakest phosphorylation), in order to illustrate, as if in an anticipation, which alcohol will be most efficiently phosphorylated, when many of them would compete for the same inorganic and organic resources in the same location, i.e., inorganic phosphates and condensing agents, respectively. This is the order that we wish to present to the reader, rather than following the organic chemist's logics of functional groups chemistry.

- Following the above comment, just the NMR of urea assisted phosphorylation is shown. I understand that all NMR will be a huge file, but at the end of each section, the authors (in just a figure) can stack the ^1H and the ^{31}P NMR, where shows the comparison of glycerol + P_i and the condensing agents, or glycerol + urea + the P sources, and so on.

3.5 Indeed, this would be a huge file of a very limited information content for the average reader of the Supplementary Information. Especially the ^1H NMR spectra are not of any good use, there is no need to show the lack of usefulness of ^1H NMR spectra in such a repetitive way, it would only dilute the information content and make the Supplementary Information document even longer, it contains 253 pages already, including the Appendix.

- Basically, all the data of the rest of the reactions (other P sources, no any data of the liquidisers) are in tables or bar graphics, it would be interesting to see the NMR (as mentioned above).

3.6 Therefore, we offer a hopefully useful compromise. We have generated seven PDF files that all contain the ^{31}P $\{^1\text{H}\}$ NMR spectra of the experiments that delivered the bar graphics of the most "successful" starting alcohols, glycerol and MPG. Almost all of these were measured in $\text{DMSO-}d_6$ (with the exception of a few glycerol + thiophosphate mixtures that were registered in D_2O as well). Since it is this solvent that produced spectra of sometimes limiting quality (usually those that contained MPG) occasionally showing a low signal-to-noise ratio (SNR), we thought we could meet the critical Supplementary Information reader somewhere in the middle.

These 7 PDF files are named according to the corresponding bar graph figure:

<Suppl. Fig. 57> (glycerol at 115°C and 75°C),

<Suppl. Fig. 61> (MPG at 115°C),

<Suppl. Fig. 62> (MPG at 75°C),

<Suppl. Fig. 71> (MPG + **cTMP**),

<Suppl. Fig. 77> (glycerol + **SP_i** at 115°C),

<Suppl. Fig. 78> (glycerol + **SP_i** at 75°C),

<Suppl. Fig. 85> (glycerol/MPG + minerals).

Each of the PDF documents contain the ^{31}P $\{^1\text{H}\}$ NMR spectra at full spectral width and an insert showing a zoom in the acyclic area around the inorganic orthophosphate, in the same order as they appear in the corresponding bar graph. Each ^{31}P $\{^1\text{H}\}$ NMR spectrum is labelled with two SNR values as determined by the MNOva software (the upper limit is 10,000), viz., that of the residual P_i peak and that of the highest peak, usually the acyclic monophosphate linked to a primary alcohol function.

In addition to the Supplementary Information 253-pages document, we uploaded these 7 PDF multi-page documents as additional separate Supplementary Dataset files, and hope that the critical reader can be satisfied with this solution, without showing the ^{31}P $\{^1\text{H}\}$ NMR spectra of all nucleotide mixtures, of which many typical examples are present in the Supplementary Information already — they were taken in

H₂O/D₂O and produced often higher quality ³¹P {¹H} NMR spectra than those of MPG phosphate mixtures in DMSO-*d*₆ — and without all monool spectra.

- The major concern of the manuscript is the quantification for the conversion % (based on the main paper: “the phosphorylation yields of 6 (measured as P_i consumption)”.

3.7 Oops, thank you! In the revised version the word “yield” has been deleted: “... the phosphorylation of 6 (measured as P_i consumption) dropped to ...”

It is well known that ³¹P qNMR is a valuable analytical technique for evaluating some phosphorus-containing compounds. But I did not find any standard (internal or external) that the authors used for the quantification.

3.8 Before we started with this work, we have evaluated all common standards (ammonium orthophosphate, triphenylphosphate, tris-(2-chloroethyl) phosphate, carboxymethylphosphonic acid, hexamethyl phosphoric acid triamide) and decided, given the extremely rich and highly dispersed signals over a broad ppm range, to quantify in terms of a reference focusing on the inorganic phosphate source. The most important reason for this is the geochemical framework/context of the study. As we wrote in the introduction of the main paper, and referring to ref. 4 being an important review by Walton et al. in *Nature Geosci.* **16**, 399-409 (2023): the main geochemical source of phosphate for the emergence of protometabolic reaction networks containing phosphate most likely were prebiotically available (soluble) simple organic phosphate esters (such as the products of our neat phosphorylation conditions). Therefore, we reference our quantification to the conversion of the inorganic phosphate source to any organic phosphate-containing products, independently of the conversion of the starting alcohol that could also react to not phosphorylated side-products, that were of no immediate use as an organic phosphate source. The organic phosphates were the “primary” phosphate source for biochemistry to evolve, see also responses 3.9 – 3.11.

- They said that for qNMR 60 s of delay was used, how many scans they used? Some figures of the ³¹P NMR show a high signal:noise ratio. This inconvenience would interfere with the quantification.

3.9 128 scans, see changes and comments on the SNR in point 3.11 (below).

- There are several references that use ³¹P qNMR for quantification (see these two: *Anal. Chem.* 2024, 96, 11198–11204 and *Appl. Sci.* 2025, 15, 323), and as it can be seen the use of a standard is almost “a must”. If the authors did not use a standard, that implies that they cannot claim “yields”, and if they just mentioned conversion % of P_i (or the others P sources) that is a mistake as well.

3.10 We did not claim yields anywhere, only inadvertently slipped this word and deleted it, thank you. We respectfully disagree with the opinion that “conversion % of P_i [...] is a mistake as well”. We have now more clearly defined in the Supplementary Information what we mean by “Conversion” throughout the paper (main and SI).

- Some explanations for these comments:

3.11 We thank Reviewer #3 for raising these points. In order to appropriately answer and clarify all questions raised above (responses 3.6, and 3.8-3.10), we have much extended the sub-section 1.2. of the revised Supplementary Information document and renamed its heading to (new):

1.2. NMR equipment, signal description, and quantification methods:

NMR spectra were recorded in CDCl₃, fully deuterated dimethyl sulfoxide (DMSO-*d*₆) and D₂O on *Bruker Avance 300, 400 and 500* spectrometers equipped with a standard BBFO probe. ¹H were recorded with 2 sec relaxation time and excitation sculpting when presaturation was needed. ³¹P nuclei through qNMR spectra were performed using 128 scans, inverse gated decoupling with 90-degree pulses and 60 sec relaxation time (longer than measured on SP_i having the slowest relaxation time) to ensure quantitative measurement of all ³¹P nuclei. ³¹P-¹H HMBC spectra were run with the standard pulse program from *Bruker*, using 512 increments with 100 ppm spectral width in F1, 4 scans and 1.6 sec relaxation delay giving 1 h experiment time. ¹⁵N experiments on

enriched compounds were run using a 90 degrees single pulse, with 30 sec relaxation delay. Additional $^{31}\text{P}\{^{13}\text{C},^1\text{H}\}$, $^{13}\text{C}\{^{31}\text{P},^1\text{H}\}$ JMOD NMR, together with ^{31}P - ^{13}C and ^{31}P - ^{15}N HMBC spectra, were taken on a *Bruker Avance NEO 600* MHz spectrometer equipped with a quadruple resonance (^1H , ^{13}C , ^{31}P , ^{15}N) inverse cryoprobe (QCI 5 mm) at the *Institut de Chimie de Toulouse, Université Paul Sabatier*.

Chemical shifts of solvents (CDCl_3 : $\delta_{\text{H}} = 7.26$ and $\delta_{\text{C}} = 77.23$ ppm; $\text{DMSO-}d_6$: $\delta_{\text{H}} = 2.50$ ppm (quintet, $J_{\text{H,D}} = 1.9$ Hz) and $\delta_{\text{C}} = 40.0$ ppm (septet, $J_{\text{C,D}} = 32$ Hz); D_2O : $\delta_{\text{H}} = 4.65$) served as internal references. Signal shapes and multiplicities are abbreviated as br (broad), s (singlet), d (doublet), t (triplet), q (quartet), quint (quintet) and m (multiplet). Where possible, a scalar coupling constant J is given in Hertz (Hz). ^{31}P NMR chemical shifts were referenced to NaH_2PO_4 ($\delta_{\text{P}} = 0.00$ ppm). Although ^{13}C NMR chemical shifts were usually referenced to DMSO ($\delta_{\text{C}} = 40.0$ ppm), in the case of spectra taken in $\text{H}_2\text{O}/\text{D}_2\text{O}$ the signal of urea (**2a**) at $\delta_{\text{C}} = 160.50$ ppm was considered as an internal reference.

The quantification of products was tested and performed using 1D qNMR methods with ^1H and ^{31}P nuclei. The quantities of the main groups of phosphorylated products were estimated using relative quantification, compared to the signal of inorganic phosphate by $^{31}\text{P}\{^1\text{H}\}$ qNMR. In most experiments, phosphorylated organic molecules were produced as a result of synthesis and were not initially present in the mixture. Therefore, the quantity of inorganic phosphate at the end of the reaction corresponded to its unreacted leftovers (if present). In conclusion, the relative quantities of organic molecules resulted from the conversion of inorganic phosphate into them.

The regions of interest were integrated and compared to the integral of inorganic phosphate, with the total sum set to 100 %. Due to the large number of different experiments, we decided not to use an internal standard and instead relied on a relative quantification technique. This approach facilitated the comparison of different experimental conditions and the ratios of initial reactants.

The results of this quantification obtained from $^{31}\text{P}\{^1\text{H}\}$ NMR spectra included only phosphorylated organic products and do not contain all other possible by-products of the reaction. Multiplicity analysis revealed that the majority of the signals appeared as singlets in $^{31}\text{P}\{^1\text{H}\}$ NMR spectra, thus, each corresponded to a single molecule available for integration, except for di-, tri-, and polyphosphate products. In other words, we define throughout this work "Conversion % P_i " as the percentage of the inorganic phosphorous source P_i , SP_i , PP_i (canaphite analogue) or **cTMP** transformed into organic phosphates of any kind (acyclic phosphate monoesters and diesters, cyclic diesters, organic diphosphates and triphosphates etc). In the time-dependent studies of the urea-assisted phosphorylation of glycerol (**5**) and MPG (**6**) we name this conversion " **P_i consumption**" to distinguish it from the degradation of urea and the formation and degradation of carbamates and cyclic carbonates as detected by ^{13}C NMR spectroscopy.

The signal-to-noise ratio (SNR) of qNMR spectra was systematically evaluated and measured using the MNova software for experiments of the kinetic study of urea-assisted phosphorylations, as well as all other ^{31}P qNMR of the phosphorylation of **5** and **6** (Sections 7.1-7.4). The kinetics results are presented in Section 5 and were incorporated in the corresponding calculations, figures, and tables. Because of the large amount of data and number of signals in the qNMR spectra concerning Sections 7.1-7.4, there the SNR of the largest signal ($< 10,000$) and that of P_i were added as Supplementary Dataset files named after the corresponding figure numbers. Whenever this SNR was very low for the lipophilic samples due to aggregation or phase separation issues, we noted it in the supplementary text.

With the exception of the mixtures containing dodecan-1-ol (**9**, Section 7.9), all other ^1H NMR spectra appeared to be too complex for their reliable interpretation and quantification of the components by qNMR. In the kinetic study (Section 5), the relative quantities of glyceryl carbamates, cyclic carbonates, and other reaction by-products were evaluated based on the urea consumption, the ^{13}C NMR signal of which was never crowded or overlapped by others, and the isotopologs could be easily deconvoluted. This approach, however, was difficult or impossible to apply when calculating conversions relative to any starting alcohol other than **9**, because of overlapping signals in the region of interest of their, both, ^{13}C or ^1H NMR signals.

- Figure S16 shows the ^1H NMR spectrum (500 MHz) of the $\text{DMSO-}d_6$ extract of a mixture of glycerol (**5**), natural isotope abundance-urea (**2a**) and NaH_2PO_4 (P_i) (1:1:1, 0.5 mmol each) after heating it neat for 120 h at 115°C , these are the net conditions for most of the experiments. Immediately, it can be seen the amount of

unreacted glycerol but based on the figure 2 (main paper) and the table S26 (same conditions) the total conversion of Pi to phosphorylated organic products (glycerol in this case) was 91%.

3.12 On larger reaction scales we observed up to 98 % P_i conversion, see Supplementary Tables 11 and 19 (for glycerol and, resp., MPG on different reaction scales). In order to avoid this misunderstanding of the values in Fig. 2 we added in its legend (**new**):

Fig. 2. Maximum percent conversion of P_i, cTMP, PP_i and SP_i after 5 days at 115°C in neat conditions. Coloured boxes: **Highest % P_i conversion** (from ³¹P NMR peak integration) to phospholipid precursors **from** alcohols **5, 6** (R = palmitoyl) and **7-10** blended with phosphate source (black circles or ovals) and dehydrating agent **1 or 2** (red stars), 1 mol equivalent each, **all measured reaction scales included, maximum values for each class of compounds (acyclic, cyclic 1,2 and cyclic 1,3)**. Pie charts (right): Summary of nucleotide distribution (larger) and nucleobase identity (smaller) after heating the neat ribonucleosides (N) at 115°C, mol ratio **11:12:13:14:2:P_i 0.25:0.25:0.25:0.25:4:1** and 1 eq **3a** as liquidiser (from signal integration of UV_{260 nm} using HPLC-HRMS, see Supplementary Figs. 129-130).

- If they try to use the deconvolution for the 1H NMR of the reaction and obtain the conversion % of glycerol-to-glycerol phosphates the difference will be huge.

3.13 This deconvolution of ¹H NMR signals does not reliably work on complex glycerol reaction mixtures; it does not give reliable values for any mixture other than those containing dodecanol (**9**) as the starting alcohol, especially when more and more products are created in the mixture, as can be seen in the stack of ¹H NMR spectra in Supplementary Fig. 44. Only the values for dodecanol conversions to other dodecanol derivatives (phosphates, carbamates in Supplementary Figs. 164-165) could be determined by ¹H NMR spectroscopy and given in the SI. We cannot reference the measured conversions to any other alcohol by ¹H NMR spectroscopy, or else for MPG (**6**) by HPLC-ELSD (Suppl. Fig. 63).

To make this clearer, we have mentioned the exception for **9** at the end of the extended sub-section 1.2 (see above, point 3.11) and we show a revised Supplementary Fig. 16, in which we have added an insert showing the higher-field ¹H NMR signals of this extract, we labelled the signals where unreacted glycerol is seen (Ha, Hb, Hx) and added the structure of glycerol (with the same labels). We added to the legend the following text (**new**):

Supplementary Fig. 1. ¹H NMR spectrum (500 MHz) of the DMSO-*d*₆ extract of a mixture of glycerol (**5**), natural isotope abundance-urea (**2a**) and NaH₂PO₄ (**P_i**) (1:1:1, 0.5 mmol each) after heating it neat for 120 h at 115 °C. **Framed insert:** zoom of the mainly not phosphorylated spectral region $\delta_{\text{H}} = 3.20 - 3.60$ ppm shows more than the ABX system (*dd*_{Ha}, *dd*_{Hb}, *tt*_{Hx}) of the unreacted starting glycerol (**5**) only.

- I don't get it if the 91% of the P_i was transformed into the glycerol phosphates why in the 1H NMR the integrals of the glycerol (starting material) are much bigger than the glycerol phosphates?

3.14 Because, in an equimolar mixture, P_i reacts very quickly with glycerol, and not much slower with glyceryl phosphates to give still higher phosphorylated glyceryl and diglyceryl derivatives as well, this before more glycerol can react with the P_i. In addition/instead, some of the glycerol reacts to not phosphorylated glyceryl side-products that obtrude the ¹H NMR spectrum and make it difficult to deconvolute the residual unreacted glycerol resonances. We modified the text under Supplementary Fig. 16 accordingly (**new**):

According to 2D spectral analyses (Suppl. Figs. 18-20), the ¹H resonances at $\delta_{\text{H}} = 3.25-3.55$ ppm show protons from the unreacted glycerol (*dd* δ_{Ha} , *dd* δ_{Hb} , *tt* δ_{Hx}) as well as other not phosphorylated glycerol derivatives. These not phosphorylated, charge-neutral side-products cannot be detected in mass spectra of mixtures that contain many different ionic phosphate esters (cf. Suppl. Figs. 27 and 28).

- Something similar happened with the nucleosides. Different from the glycerol phosphates, the nucleotides are well known signals in the 1H NMR. For quantification in nucleoside phosphorylation is more realistic the use, for instance, for the purines the anomeric proton. For adenosine the chemical shift of the anomeric proton for the adenosine, 2'-AMP, 3'-AMP, 5'-AMP and the 2',3'-cAMP are well known (see: J. Am. Chem. Soc. 2023, 145, 23781–23793) so the quantification would be more accurate.

3.15 To reliably quantify without any (or minimal) doubt the unreacted nucleosides in the ^1H NMR spectra was impossible as well. Too many different side-products obtrude the clear identification of the unreacted nucleosides, especially in the mixtures containing all four. We have tried several times and we convinced ourselves not to skate on such thin ice. It was much more reliable to reference the quantities of unreacted P_i in the ^{31}P $\{^1\text{H}\}$ NMR spectra, the more so as the NMR samples of nucleotide mixtures were taken in water, thus, no P_i solubility issues, aggregation or phase separation problems occurred that could have produced a too low SNR. As explained in comment 3.8, in our opinion, to quantify P_i conversion to the nucleotides is in the context of this study more important than to know the yields for each nucleoside-to-nucleotide phosphorylation. The strongest interest in phosphorylating ribonucleosides is the formation of oligonucleotides anyway. We have concentrated our efforts to find higher oligomers by ^{31}P DOSY, for instance, and HPLC-MS.

From an analytical point of view, it was for us very important that we systematically quantified the phosphorylation of all starting alcohols in exactly the same way — except dodecanol **9**, that was not well phosphorylated anyway. We wanted to avoid comparisons of the incomparable, like P_i conversions in glycerol and MPG mixtures with nucleoside conversions in nucleotide mixtures (with ...? in nucleotide-glycerol mixtures).

- One example of these discrepancies of quantification is figure S62. According to this figure, the authors claimed: “It seemed clear that MPG poorly reacted in the presence of cyanamide (65% unreacted MPG left after 5 days) when compared to urea (18% left after 5 days)” these % of conversions were measured using RP-HPLC and analyzed by ELSD, but table S27 shows that MPG phosphorylation using 1 eq of cyanamide or 1 eq of urea was 51.5% and 52.9%, respectively.

3.16 We added to this part, after the Supplementary Fig. (now) 63, the following text:

The results from HPLC differ slightly from those obtained by NMR in terms of the conversion of the starting molecule. The NMR analysis was based on P_i , showing only the conversion of inorganic phosphate into organic phosphorylated products. However, this is different from the conversion of MPG into all other phosphorylated and not phosphorylated reaction products.

Another reason for the variation may lie in the analytical approach. While the NMR spectrum includes all molecules present in detectable quantities, HPLC depends on the type of chromatographic column used for separating reaction products. In the experiment, we used a C18 column, which is well-suited for long-chain lipids. However, it does not retain small polar glyceryl phosphates formed due to the deacylation of MPG, leading to differences in the determined quantities.

The same plot shows that increasing the eq of urea the % increased but not mentioned at all what happens if the eq of cyanamide increases as well.

3.17 For some reason we did not carry out this experiment... A leading thought for the experiments with cyanamide, or their lack, was that cyanamide will partly react with the residual amount of water, for example, from humidity, and will produce urea in an uncontrolled way, which makes it difficult to dissect the reactivity of urea from that of cyanamide. See also responses 3.18 and 3.2.

- Urea as an “activator”, promotor or just as a condensing agent with inorganic phosphate is very well known and has a history in prebiotic phosphorylation even has been intensely investigated. Many aspects for consideration:

- Based on the results presented reactions (e.g. Fig. S57, S60, S66, S82), cyanamide gave better conversion % than urea, so why don't pursue this agent that is less known? In the case of Struvite as a P source the difference was more than 20%, just for comparison.

3.18 Cyanamide is just as well-known as urea in its plausible rôle as a prebiotic condensing agent. The first-generation prebiotic chemists such as Orgel, Oró and others (Schimpl & Lemmon) have pioneered cyanamide. In our opinion, and because of the much higher chemical potential of cyanamide when compared to urea (cf. Fig. 1), cyanamide in the presence of water vapour (humid periods in the Hadean night time) or liquid water, even if only episodically wetted, hydrolysed to urea over geochemically

relevant time periods, whereas urea is much more resistant towards hydrolysis and heat (see also response 3.2). This is the reason for why we studied urea- P_i blends much more in detail than cyanamide- P_i blends.

From a thermodynamic point of view, it is not at all astonishing that cyanamide can give higher phosphorylation yields, when appropriate lab conditions are set up. We wanted to relate better to a plausible geochemical scenario.

- There is not any NMR (for supporting the claims) of the blank reactions (blank means just the alcohol or nucleoside and the P source, then the P source and the “Dehydrating /Condensing Agents/Liquidisers/Catalysts). There are some experiments of urea (heating to see the degradation) but not with the P source.

3.19 We did show NMR evidence for the ‘blank experiments’ regarding P sources for the reaction of glycerol + P_i (Suppl. Fig. 64, in the unrevised SI it was Fig. S63), glycerol + SP_i (Suppl. Fig. 73, in the unrevised SI it was Fig. S70), and BTG (7) + P_i (Suppl. Figs. 149-150, in the unrevised SI Figs. S146-147). However, we missed **cTMP**, thank you! We have repeated this ‘blank’ **cTMP** + glycerol experiment at 115 °C and could confirm the earlier results. We now have added more figures, i.e., for the “blank” phosphorylation of glycerol with **cTMP** (Suppl. Fig. 66) and we realised that we have discovered something new. We have therefore added the following text (**new**) and Supplementary figures:

In the main paper: With **cTMP** at 115 °C, but not 75 °C, **5** became partly carbonylated even in the absence of **2a** or **1**, which meant that CO_2 from the air could be activated by **cTMP** at 115 °C and transferred to **5**, its glyceryl phosphate derivatives or **6**, to appear as 1,2-cyclic carbonates (cf. Supplementary Fig. 69).

In the Supplementary Information: The use of equimolar amounts of **cTMP** with glycerol (**5**) gave excellent conversions to organic phosphates (71.6-98.2 %) at both temperatures 75 and 115 °C (Suppl. Fig. 68) even in the absence of a condensing agent. Surprisingly, heating **5** with **cTMP** and **no** added **urea** in open vessels at 115 °C provoked the formation of cyclic carbonates of diglyceryl phosphodiester **diGI(CO)₂P** and **diGICOP** (but not at 75 °C, see Suppl. Fig. 66), viz. exactly the same molecules (cf. Suppl. Fig. 25B) that are being generated after 8 hours of heating **5** in the presence of P_i and urea (**2a** and **2b**, see Suppl. Figs. 45, 46 and Suppl. Table 8). Since urea was lacking in this **cTMP** experiment, the only other source of carbonyl groups could have been carbon dioxide from the air. It appears that CO_2 was activated by 115 °C-hot **cTMP** and then the carbonyl group transferred to the glyceryl phosphates (Suppl. Fig. 69).

The presence of a condensing agent did not increase the total amount of organic phosphates very much (+ 3-8%, Suppl. Table 30), but more **diGI(CO)₂P**, **diGICOP** and cyclic phosphates were detected in **cTMP** reactions with cyanamide (**1**) and urea (**2a**), see Suppl. Fig. 67. This confirms that the additional dehydrating power provided by **1** or **2a** is used to carbamylate and then cyclise more acyclic glyceryl monophosphates into cyclic carbonates and cyclic phosphates. Owing to the fact that we registered the presence of P_i (0 ppm) and PP_i (-9.29 ppm) but hardly ever **cTMP** itself, we can conclude that the reaction conditions are always favourable for a ring-opening of **cTMP**.

The fact that **cTMP** appeared to bind CO_2 in our neat 115 °C-hot conditions, and transfer a carbonyl group to 1,2-diols (phosphorylated or not = P/OH), is consistent with the work of others who discovered the capacity of aqueous trisodium orthophosphate to bind CO_2 (suppl. ref 33). We can hypothesise that neat hot **cTMP** at 115 °C activates CO_2 through a mixture of carboxylation (Suppl. Fig. 69, horizontal sequence) and oligomerisation of **cTMP** (vertical steps) to produce bridged through carbonyl groups (marked red) oligomers that have a $(cTMPCOcTMP)_n$ scaffold bearing varying degrees of peripheral carboxylation. The heat-driven carboxylation-oligomerisation process of dry **cTMP** would be accompanied by the elimination of sodium oxide from the trisodium salt of **cTMP**. The oligomeric bridging carbonyl groups could then react with nucleophiles such as 1,2-diols, whereby highly basic Na_2O would bind their protons to give water that ring-opens **cTMP** units (ring-opened PPP_i , PP_i and P_i not shown in the figure) while glycerol (**5**) (or MPG (**6**)) is phosphorylated and/or carbonylated : see Section 7.4.2 in the SI.

We have also re-quantified all aforementioned **cTMP** + glycerol ^{31}P { 1H } NMR results shown in Supplementary Table 30, in order to separate the integrals of the ^{31}P NMR resonances, thus, separately quantify the carbonylated diglyceryl phosphodiester **diGI(CO)₂P** and **diGICOP**, and **added** (this **new**) information to its caption:

Supplementary Table 30. Data to Supplementary Fig. 68. Calculation of total conversion (conversion of initial amount of **cTMP** to phosphorylated organic products) was made by subtraction of inorganic compound quantities (**P_i**, **PP_i** and **cTMP**) from the sum of all integrated ³¹P{¹H} NMR peak areas (in DMSO-*d*₆). **GIP** comprise all acyclic organic phosphate peaks except those for **diGI(CO)₂P** and **diGICOP** (Supplementary Figs. 66 and 67). [See modified Supplementary Table 30 in the revised SI].

Just in case Reviewer #3 wishes to ask whether we have been able to confirm this hypothesis (that cTMP was carbonylated at 115°C) with ¹³C-labelled CO₂: No, we haven't and we won't be able to carry out such extended late experiments anymore, not for this paper. This is why we call it hypothetical and put this word in the supplementary text and figure legend of Supplementary Fig. 69. Maybe in some future work somebody will want to carry our ¹³C-labelled control experiments and publish the results in a separate paper.

- Some contradictory results. Example: Glycerol + P_i (no urea) conversion 35% and 3%, 115 °C and 75 °C, respectively. But then in the conclusion the claim was: “glycerol was phosphorylated even in the absence of any urea showing P_i conversions 36-77 % after 4-5 days at 115 °C”. Even worse, with cTMP without urea glycerol was phosphorylated about 90% (based on table S30), so why even bother to use urea or cyanamide if just glycerol and cTMP worked?

3.20 Again, because we were not just looking for best yields, we wished to compare different prebiotic reagents under the same model conditions (without necessarily going as deep into the mechanisms as for urea-assisted phosphorylations), see also our response 2.5 to Reviewer #2. We did mention in the SI that **cTMP** was a questionable prebiotic P source because of its very high chemical potential and its lability to hydrolyse when the water became more acidic owing to huge amounts of CO₂ in the Hadean atmosphere.

- MPG required the presence of urea to become efficiently phosphorylated at 115 °C, without urea at most 10% P_i was converted to organophosphates, but then (not with P_i but with cTMP the conversion of MPG without urea was 76% based on figure S68. Same comment, no need of urea if only MPG and cTMP worked.

3.21 See response 3.20 above.

- Section 5.1.2. It says: “The presence of equimolar amounts of urea (2a:2b), P_i and glycerol (5) in a neat mixture that was heated at 115 °C not only gave clear evidence for the accelerated degradation of urea to ammonium cyanate within the same timeframe” but based on the figure S42a after 24 h was clear the degradation, so why on the phosphorylation reactions the experiments were for 5 days?

3.22 Because we did not know right from the start how long these neat, dry reactions would take. In observing the mixtures for such a long time period, we were able to discover the formation of more complicated late products, such as the carbonylated diglyceryl phosphodiester (**diGI(CO)₂P** and **diGICOP**). This in turn led us to the conviction that carbamoylated and carbonylated products must have had an advantage in being more resistant towards degradation, they better persisted the Hadean weather.

In the revised SI we have added at the end of Section 2.1 **CarouselTM reactions** the following explicative text (**new**):

The time scale of 5 days was chosen to capture the formation of all main phosphorylation products along with by-products of urea degradation. The extended duration allowed us to track the accumulation and degradation of many late products beyond the degradation of urea and complete consumption of the initial inorganic phosphate source.

- If the urea's degradation is clear after 24 h to ammonium cyanate, it would be a good idea to show the reaction e.g. glycerol + ammonium cyanate + P_i under the same conditions. This experiment would show the role of urea or the role of ammonium cyanate in the assisted-phosphorylation reaction.

3.23 Indeed, this experiment could have been done. After having tried very hard, but in vain, to find proof in the [¹³C]urea-P_i blend with glycerol for the at least transient presence of [¹³C]CP_i or [¹³C]cyanate after 2 hours of heating, and having had the proof of cyanate formation through ¹⁵N exchange in situ in the neat mixtures under the actual reaction conditions, we were convinced enough, we just let it go. Getting yet another indication (not proof) by artificially adding ammonium cyanate seems unnecessary in the face of

the ¹⁵N exchange, because replacing half the natural urea with [¹⁵N₂]urea is less invasive than replacing all the urea with ammonium cyanate, and knowing that ammonium ions strongly accelerate urea-assisted phosphorylation.

From a geochemical point of view, ammonium cyanate isomerises through the Wöhler reaction to produce urea, so it would be really beating it out.

Minor concerns:

- Figure 2 is not completely understandable. Orange exemplifies the Total conversion %, but only shows cyanamide and Pi and Struvite, while urea and cTMP, PPi and SPi results. Next, for the acyclic phosphates %, shows cyanamide and cTMP and Struvite, while urea and Pi, PPi and SPi. For the Cyclic 1,2-phosphates % cyanamide and Pi and Struvite, while urea with the rest. Finally, is it complex to see the full product distribution. The best idea will be divided into two portions the cyanamide and urea results, each one with the corresponding P source.

3.24 We admit that the figure is highly concentrated, it only shows the highest conversions, in order to give the impression of how far one can get in neat hot mixtures with whatever tested P source and activating agent. We hope that the modified figure legend helps understanding the figure, see response 3.12.

- Following the above, taking as an example glycerol: The data were taken from figure 2, so when the hyphen is there, it is because based on the figure that data is not shown, so it is complex to see the full total conversion. Also, the first box started with cTMP, next box (yellow one) started with Pi. There is no sequence for reading the data. It is not necessary to change completely the figure, just redraw in a way that will be easier to understand and to see the conversions.

Total Phosphorylation%	Acyclic phosphates	Cyclic 1,2-phosphates	Cyclic 1,3-phosphate
1 2 1 2 1 2 1 2			
Pi 97 - - 81 8 - - 4			
cTMP - 98 77 - - 3 - 11			
PPi - 85 - 74 - 6 - 5			
SPi - 28 - 28 - 1 - -			
Struvite 64 - 58 - 2 - 1 -			

3.25 No, there is no sequence for reading the data, correct. This means, in Fig. 2 one cannot simply add up the conversion values of acyclic, 5-membered ring products and 6-membered ring products to obtain the total conversion values, because it only shows the highest conversions for each class. See also response 3.12.

If somebody wants to see which conditions favour a particular class of compounds (such as, for example, monoacyl-1,2-cyclic phosphates, that have been shown by Krishnamurthy & Sutherland to be able to form vesicles, they can find the most effective neat, hot conditions quite easily in Fig. 2. The otherwise interested reader is encouraged to find it the Supplementary Information. We have provided many internal links in the SI, so the interested reader can jump directly from the Table of Contents to the topic of interest.

- In the same figure, the part of the nucleosides, if the authors mixed A, G, U and C (the canonical nucleosides), what does the other (2%)? What another nucleoside can be?

3.26 As mentioned in Section 7.5.8. Supplementary conclusions on nucleoside phosphorylation, these are inosylate, N⁶-carbamoyladenine, O²,2'-cyclouridine, etc. We added at the end of the legend of Fig. 2 (new): Other = non-canonical nucleobases and nucleos(t)ides, see Supplementary Figs. 129-130.

- Why they used the terms “Heated neat blends”?

3.27 Because we blended neat urea, neat sodium dihydrogen phosphate and neat glycerol, blended them to make the mixture as homogeneous as possible, and then heated the blended mixture.

- Table S19 shows the reaction scale dependence of conversion of MPG, how they explained that using [13C]urea was a 100% of conversion? But when they used [15N2]urea was only 29% of conversion.

3.28 We have added to the caption of Supplementary Table 19 (new): There is a fluctuation in the conversion of P_i in the series of experiments with identical reaction scale, which may be related to the solid state of the crude mixture, the use of starting molecules from different batches (MPG, labelled and unlabelled urea and P_i), and the generally low solubility of the mixture after heating.

- “At the 0.5 mmol scale we detected that P_i consumption reached an end after 4 hours showing an exponential rate constant at 115 °C that was essentially the same as in the glycerol phosphorylation reaction”, again why let the reaction for 5 days if after 4 hours all the P_i was consumed?

3.29 See response 3.22 above.

- Why in SI section 2.2 explained the procedure, showed the Fig S12, but then refer the Fig S54 – S56, why didn't show after this section?

3.30 Because Suppl. Fig. 12 shows description of methods and Suppl. Figs. 54-56 the discussion of results.

- Figure S19a, can the authors explain why in the ^{31}P NMR around 16 ppm there are at least 4 signals?

3.31 These signals show different 5-membered-ring cyclic phosphates, at least two of them are shown in Suppl. Figure 27 D. Others could be minor 5pGlp and/or 5pGlpG15P, or maybe a carbamate or carbonate of 5pGl.

- Figure 1 shows Dehydrating / Condensing Agents/ Liquidisers / Catalysts, why the use of catalysts word?

3.32 We have added a star * next to “catalysts” and in the legend (new): * Urea^{ref. 43}, formamide and other carboxamides have been claimed in the literature to act as possible organo-catalysts in the phosphorylation of alcohols, cf. Fig. 4A and Supplementary Information (section 3.1).

- Only was used thymidine (for deoxyribonucleosides series), so that need to be in the figure 1.

3.33 Yes, it is, in brackets.

- Figure S59, shows 1H - $^{31}P\{^1H\}$ HMBC spectra, the X axis is not shown.

3.34 In MNova it is not possible to show four different horizontal spectra in this way of 2D representation. Therefore, we have mentioned in the figure legend (here underlined): ... 1H - $^{31}P\{^1H\}$ HMBC spectra (500 MHz 1H [horizontal axis], $\delta_H = 5.0$ - 3.0 ppm (each column), 202.5 MHz $^{31}P\{^1H\}$ [vertical axis], DMSO-*d*6) of the extract ... etc. In addition, we have added above the HMBC a stack of the 1H NMR spectra (Suppl. Fig. 59) that highlights in grey the zone that is shown in the HMBC below.

- It's a good idea to make some comments or a possible explanation of the differences in the product distribution between cyanamide and urea, example, why cyanamide sometimes gave more of the cyclic products than the acyclic ones?

3.35 It really depends on the experiment. For example, in experiments with **cTMP** there are more cyclic products in the experiment with urea, than with cyanamide. We feel that it must suffice to state that cyclic phosphates need one equivalent more of any condensing agent, and to show, for instance, in Fig. 4C a reaction cascade where carbamoyl groups serve to cyclise into 1,2-cyclic carbonates. In addition, we have added the text (shown in response 3.19): This confirms that the additional dehydrating power provided by **1** or **2a** is used to carbamylate and then cyclise more acyclic glycerol monophosphates into cyclic carbonates and cyclic phosphates.

- The figures S63 and S65 only show a window of the complete spectra, it would be good if the authors show the spectra from 20 to -25 ppm, since the signal of cTMP is around -22 ppm.

3.36 Good point, done: 20 to -30 ppm (in now Supplementary Figs. 66-68).

- The authors have proof of formation of the compound on figure S85, the 2',3'-cyclic adenosine monophosphate with the urea in the 5'- position?

3.37 Good observation. No, this was a copy-paste error. We do have (cumulative) proof of the favoured formation of *N*⁶-adenosine-1,2-cyclic phosphate. In Suppl. Table 45-1 peak #25 corresponds to (H₂NCO)U>p; #31 to (H₂NCO)G>p and #44 possibly to a fragment of (H₂NCO)A>p, it is consistently the nucleobases that are preferentially carbamoylated. We have also found that 2',3'-cyclic phosphates are the preferred 5-membered-ring phosphates. We have corrected this structure in Suppl. Fig. 88 of the revised SI, thank you.

- In the case of adenosine and guanosine phosphorylation, the easiest way to see the products is checking the anomeric region (around 5.8 and 6.4 ppm), figure S86 shows very poor conversion to the adenosine phosphates, while figure S89, shows good signals for guanosine phosphorylation. If the authors check this 1H NMR, they can identify the 2'-, 3'-, 2',3'- and even the 5'- phosphates products.

3.38 The attempt to quantify nucleotides by ¹H NMR was hindered by the complexity of the spectrum. Cross-correlations that are easily detected by 2D ¹H-¹H/³¹P HMBC do not correspond to isolated or well-defined signals in the ¹H NMR spectrum. Additionally, we cannot exclude the presence of other signals from non-phosphorylated molecules in the same region of the ¹H NMR spectrum, that are not visible in the HMBC spectrum. Therefore, all further quantification was performed using ³¹P {¹H} NMR spectroscopic and LC-MS methods.

- For pyrimidines the easiest would be the aromatic proton of the ring core of uracil or cytosine.

3.39 The complexity of the nucleotide mixtures did not allow for a doubtlessly reliable identification of all C5 and C6 ¹H NMR resonances. We would have to confirm such interpretation through ¹H-¹H HMBC which we did not undertake.

- Figure S103 does not give any information, a good idea would be to compare the initial NMR of each individual nucleoside and then the spectra after the reaction. There are a lot of signals, and the graphic lost sense.

3.40 We like this figure anyway. The comparison between the ¹H NMR spectra of each pure nucleoside and that of the mixture would not give any information about different phosphorylated products depending on the nucleosides, whereas the comparison of ³¹P {¹H} NMR spectra do show interesting differences, nuances. For example, the absence of organic pyrophosphates for C and U, or the absence of 6-membered-ring cyclic phosphates for T, G and A, or the absence of dinucleotides for T, etc. To compare the initial spectra might have sense by ¹H NMR, but what information would we expect to obtain?

- It is confusing that figure S100 shows adenosine conversion 35%, but under the same conditions (urea and 1 eq 3a, 115 °C for 5 days), figure S122 shows almost 91% of conversion, how happened this?

3.41 While the first summarises the products of an equimolar mixture, the second shows a four-fold excess of urea, which expectedly gives a higher P_i conversion and apparently produced more organic pyrophosphate denoted Npp (presumably, adenosine-5'-diphosphate).

- Figure S132 and the corresponding table S49 show the quantification of the reaction of the mixture of nucleosides + urea + P_i and 3a. There are a lot of signals, the quantification based on that figure is almost impossible. The data in my opinion is better just to show the NMR but not make any claim about the conversion.

3.42 Quite on the contrary, the ppm ranges for each group (from left to right: 5-membered-ring cyclic, acyclic monoester, residual inorganic phosphate, acyclic diester, 6-membered-ring cyclic) are very different and easily distinguishable, as indicated in the legend. Like for the other nucleotide mixtures, these ppm regions served to separate and integrate each group region without superimposed resonances. The P_i signal at 0 ppm could easily be found and cleanly integrated, that way, the P_i conversion was precisely determined. The interpretation of all these regions is in accord with the expected order of their diffusion rates as shown in the DOSY containing the same mixture of phosphorylated ribonucleosides but without added valine (see Suppl. Fig. 128).

- Something that is not clear why the authors chose valine? Why the relevance of this amino acid and not another one? Why only use valine for the nucleoside phosphorylation and not for the alcohols?

3.43 Valine is an unquestioned prebiotic α -amino acid, found again and again in many prebiotic model experiments, as a racemic mixture, of course. Valine was also found, quite recently, in brought-home Bennu powder (<https://doi.org/10.1038/s41550-024-02472-9>). It was an easy choice, and a lucky strike, too! We chose valine for the nucleosides because the commercial nucleosides were enantiomerically pure. The other alcohols were either achiral or racemic, so how else could we expect to measure a diastereoisomeric difference between L-valine and D-valine? See also the last two sentences in our response 2.5 to Reviewer #2.

- The preliminary data of the vesicles are good, but I think that they can get deeper in these experiments and maybe try to write another manuscript. There is not a lot of information about this on the main paper, so it would be good to separate.

3.44 Alas! This is one of the last experimental papers of the corresponding author. We wish to publish now these preliminary, but very important experimental outcome on the vesiculation of MPG mixtures. Better now than maybe never.

- Overall, there are some interesting ideas presented herein; nevertheless, the thoroughness of the analytical/kinetic work (SI) is not fully used in the main paper. Some experiments, ideas and conclusions that the authors wrote in the SI file would fit in the main paper.

3.45 This is a great suggestion. Following your encouragement, we shifted the (former) supplementary discussion on the reactivity of monools and pyruvic acid into the discussion of the main paper.

- I recommend these matters be reviewed and as needed, experiments repeated under more stringent conditions (like the blank reactions, as well try to see if they can quantify using 1H NMR) before the manuscript be considered for publication.

3.46 Thank you so much for your tedious, patient and meticulous work, it has served to make our paper better for more readers, we hope.